# Rethinking Evaluation Strategy for Temporal Link Prediction through Counterfactual Analysis

**Aniq Ur Rahman**[1]    **Alexander Modell**[2]    **Justin P. Coon**[1]
[1]University of Oxford, U.K.    [2]Imperial College London, U.K.
`aniq.rahman@eng.ox.ac.uk, a.modell@imperial.ac.uk, justin.coon@eng.ox.ac.uk`

## Abstract

In response to critiques of existing evaluation methods for Temporal Link Prediction (TLP) models, we propose a novel approach to verify if these models truly capture temporal patterns in the data. Our method involves a sanity check formulated as a counterfactual question: "What if a TLP model is tested on a temporally distorted version of the data instead of the real data?" Ideally, a TLP model that effectively learns temporal patterns should perform worse on temporally distorted data compared to real data. We provide an in-depth analysis of this hypothesis and introduce two data distortion techniques to assess well-known TLP models. Our contributions are threefold: (1) We introduce simple techniques to distort temporal patterns within a graph, generating temporally distorted test splits of well-known datasets for sanity checks. These distortion methods are applicable to any temporal graph dataset. (2) We perform counterfactual analysis on TLP models such as `JODIE`, `TGAT`, `TGN`, and `CAWN` to evaluate their capability in capturing temporal patterns across different datasets. (3) We propose an alternative evaluation strategy for TLP, addressing the limitations of binary classification and ranking methods, and introduce two metrics – average time difference (`ATD`) and average count difference (`ACD`) – to provide a comprehensive measure of a model's predictive performance. The code and datasets are available at: https://github.com/Aniq55/TLPCF.git.

## 1 Introduction

In static graphs, link prediction refers to the task of predicting whether an edge exists between two nodes after having observed other edges in the graph. Temporal link prediction (TLP) is a dynamic extension of link prediction wherein the task is to predict whether a link (edge) exists between any two nodes in the future based on the historical observations (Qin and Yeung, 2023). The predictive capability of TLP models make them useful in applications pertaining to dynamic graphs, such as product recommendations (Qin et al., 2024; Fan et al., 2021), social network content or account recommendation (Fan et al., 2019; Daud et al., 2020), fraud detection in financial networks (Kim et al., 2024), and resource allocation, to name a few.

In the TLP literature (Kumar et al., 2019; Trivedi et al., 2019; Xu et al., 2020; Rossi et al., 2020; Wang et al., 2020; Cong et al., 2023), the TLP task is treated as a binary classification problem where the query

$q_1$ : "Does an edge exist between the nodes $u$ and $v$ at time $t$?"

is processed by a model and then compared with the ground truth following which metrics such as area under the receiver operating characteristic curve (AU-ROC), and average precision (AP) are reported. The ground truth consists of positive samples, and a fixed number of random negative samples. There are a couple of issues in the binary classification approach. Firstly, the timestamps in the query are restricted to the timestamps present in the ground truth, which makes the evaluation

biased and does not test the model's performance in the continuous time range. Secondly, checking for the existence of an edge at a specific timestamp is an ill-posed question, and instead the existence of an edge should be queried within a finite time-interval. Lastly, the negative edge sampling strategy, and the number of negative samples per positive sample impact the performance metrics as seen in EXH (Poursafaei and Rabbany, 2023).

Alternatively, in a rank-based approach, the query is formulated as:

$$\mathsf{q}_2 : \text{"Which nodes are likely to have an edge with node } u \text{ at time } t\text{?"}$$

In this case, the model returns an ordered list of nodes arranged from most likely to least likely. Then, the rank of the ground truth edge is returned if a match is found, and if not, a high number is reported. For all the edges in the test data, metrics such as Mean Average Rank (MAR) or Mean Reciprocal Rank (MRR) can be reported to assess the performance of the model (Huang et al., 2024). While the rank-based metrics are more intuitive than AU-ROC and AP, the issues regarding binary classification mentioned above still remain unaddressed. To give a true picture of the predictive power of the TLP models, a penalty term should be introduced to account for the nodes that are incorrectly estimated to form an edge with node $u$ at time $t$.

In a recent work, Poursafaei et al. (2022) highlighted that the state-of-the-art (SoTA) performance of some TLP models on the standard benchmark datasets is near-perfect. This is counterintuitive because TLP is a challenging task, even more challenging than link prediction of static graphs, due to the additional degree of freedom in the data induced by the temporal dimension. The flaw in the evaluation method is attributed to the limited negative sampling strategy, and the authors propose a new negative edge sampling strategy which results in a different ranking of the baselines.

Inspired by the critique of the evaluation method, we propose a method to conduct sanity check of the TLP models to determine if they truly capture the temporal patterns in the data. The sanity check is formulated as the counterfactual question (Pearl, 2019):

> "What if a TLP model which is trained on a temporal graph is tested on *temporally distorted* version of the data instead of the real data?"

Ideally, a TLP model which is capable of learning the temporal patterns should perform worse on temporally distorted data compared to the real data. We conduct an in-depth analysis of this argument and introduce various data distortion techniques to assess well-known TLP models.

**Contributions**    The contributions of our work can be summarised as follows:

- We introduce simple **techniques** to distort the temporal patterns within a graph. These techniques are then used to generate temporally distorted version of the test split of some famous datasets which can be used for **sanity check**. Moreover, the distortion methods can be applied to any temporal graph dataset.

- We perform **counterfactual analysis** on TLP models such as JODIE (Kumar et al., 2019), TGAT (Xu et al., 2020), TGN (Rossi et al., 2020), and CAWN (Wang et al., 2020) to check whether they are capable of capturing the temporal patters within various datasets.

- We propose an alternative **evaluation strategy** for TLP through which the existing pitfalls of binary classification and ranking methods can be avoided. We also propose two **metrics**: average time difference (ATD), and average count difference (ACD) to measure the performance of TLP models. These metrics can provide a holistic picture of a model's predictive performance.

**Organization**    In Sec. 2, we define temporal graphs and the associated notations. We also provide a brief overview of interpreting temporal graphs as point processes, which forms the theoretical foundation of TLP. In Sec. 3, we formalize the counterfactual analysis through logical arguments, and also propose data distortion techniques. The results of the counterfactual analysis are presented in Sec. 4 along with the details of the datasets and TLP models used for evaluation. In Sec. 5, we suggest a generative evaluation approach for TLP, and discuss the broader impact and limitations of our work.

## 2 Preliminaries

### 2.1 Definitions

In TLP literature, continuous-time temporal graphs with *ephemeral edges* are often considered, where edges represent interaction events between two nodes at a specific point in time. Alternatively, temporal graphs can be defined with edges that appear at a certain time and either persist for a duration (Celikkanat et al., 2024; Farzaneh and Coon, 2023) or accumulate indefinitely. In this work, we focus on the ephemeral edge temporal graph, also known as interaction graphs (Qin et al., 2024) or unevenly sampled edge sequence (Qin and Yeung, 2023).

**Definition 2.1.** A **temporal graph** with $m \in \mathbb{N}$ ephemeral edges formed between nodes in $\mathcal{U}$ and $\mathcal{V}$ is defined as $\mathcal{G} = (\mathcal{U}, \mathcal{V}, \mathcal{E})$, where $\mathcal{E} \triangleq \{(u_i, v_i, t_i) : i \in [m], u_i \in \mathcal{U}, v_i \in \mathcal{V}, t_i \in \mathbb{R}\}$ denotes the set of edges. The tuple $(u, v, t)$ is referred to as an edge event.

While the definition caters to bipartite structure, with $\mathcal{U} = \mathcal{V}$, it can also represent general graphs.

**Definition 2.2.** The occurrences of a particular edge $(u, v)$ in $\mathcal{E}$ is denoted as $\mathcal{E}_{(u,v)}$ and defined as $\mathcal{E}_{(u,v)} \triangleq \{(u, v, t) : (u, v, t) \in \mathcal{E}\}$.

**Definition 2.3.** The slice of edges in $\mathcal{E}$ with timestamps in the range $(t_1, t_2)$ is denoted as $\mathcal{E}(t_1, t_2)$, and defined as $\mathcal{E}(t_1, t_2) \triangleq \{(u, v, t) : (u, v, t) \in \mathcal{E}, t \in (t_1, t_2)\}$.

**Definition 2.4.** The timestamps in $\mathcal{E}$ consisting of $m \in \mathbb{N}$ edges can be extracted through a function $\mathscr{T} : (\mathcal{U} \times \mathcal{V} \times \mathbb{R})^m \to \mathbb{R}^m$ as $\mathscr{T}(\mathcal{E}) \triangleq \{t : (u, v, t) \in \mathcal{E}\}$.

### 2.2 Point Process

Perry and Wolfe (2013) modelled the interaction events of a directed edge $(u, v)$ as an inhomoegenous Poisson point process. In a recent work on continuous-time representation learning on temporal graphs, Modell et al. (2024) followed suit, and assumed $\mathcal{E}_{(u,v)}$ to be sampled from an independent inhomogenous Poisson point process with intensity $\lambda_{(u,v)}(t)$. The number of edge events $(u, v)$ between timestamps $t_1$ and $t_2$ follow a Poisson distribution with rate $\int_{t_1}^{t_2} \lambda_{(u,v)}(t) \, dt$, i.e.,

$$|\mathcal{E}_{(u,v)}(t_1, t_2)| \sim \text{Poisson}\left(\int_{t_1}^{t_2} \lambda_{(u,v)}(t) \, dt\right). \tag{1}$$

To connect the present to the past, Du et al. (2016) view the intensity function $\lambda^{\star}_{(u,v)}(t)$ as a nonlinear function of the sample history, where $\star$ indicates that the function is conditioned on the history. The conditional density function for edge $(u, v)$ is written as

$$p^{\star}_{(u,v)}(t) = \lambda^{\star}_{(u,v)}(t) \exp\left(-\int_{t'}^{t} \lambda^{\star}_{(u,v)}(\tau) \, d\tau\right), \tag{2}$$

where $t' < t$ is the last time when edge $(u, v)$ was observed. The goal is to find the parameters $\lambda^{\star}_{(u,v)}(t) : 0 < t \leq T$ which can describe the observation $\mathcal{E}_{(u,v)}$. This is done by minimizing the negative log likelihood (NLL) at the timestamps of edge occurrence (Shchur et al., 2021):

$$\min_{\lambda^{\star}_{(u,v)}(t) : 0 < t \leq T} - \sum_{t \in \mathscr{T}\left(\mathcal{E}_{(u,v)}\right)} \log\left(\lambda^{\star}_{(u,v)}(t)\right) + \int_{0}^{T} \lambda^{\star}_{(u,v)}(\tau) \, d\tau, \quad T = \max \mathscr{T}\left(\mathcal{E}_{(u,v)}\right). \tag{3}$$

In (Shchur et al., 2021), the operation of a neural temporal point process is summarized as:

- The edge events in $\{(u, v, t_i) : i \in [m]\}$ are represented as feature vectors $\boldsymbol{x}_i = f_{\mathfrak{e}}(u, v, t_i)$,
- The historical feature vectors are encoded into a state vector $\boldsymbol{h}_i = f_{\mathfrak{h}}(\boldsymbol{x}_1, \cdots \boldsymbol{x}_{i-1})$,
- The distribution of $t_i$ conditioned on the past is simply conditioned on $\boldsymbol{h}_i$.

The functions $f_{\mathfrak{e}}$ and $f_{\mathfrak{h}}$, as well as the conditioning on $\boldsymbol{h}_i$, can be implemented using neural networks.

**Conjecture 2.1.** *The samples from a neural temporal point process are **learnable**, i.e., a model exists which can perform temporal link predictions based on the past observations.*

# 3 Counterfactual Analysis

**Experiment Setup**  A model $f$ is trained on a temporal graph $\mathcal{E}_{\text{train}}$ and tested on $\mathcal{E}_{\text{test}}$ through the binary classification approach resulting in metrics such as AU-ROC, and AP. In general, $\mathcal{E}_{\text{train}} = \mathcal{E}(0, \tau_0)$, and $\mathcal{E}_{\text{test}} = \mathcal{E}(\tau_0, T)$, i.e., the train and test data are chronologically split from the same temporal graph which is assumed to be generated through a common causal mechanism.

In light of the experimental setup, we ask the question: "Would the model $f$ which is trained on $\mathcal{E}_{\text{train}}$ perform well if tested on a distorted version of $\mathcal{E}_{\text{test}}$ instead of $\mathcal{E}_{\text{test}}$?" To formalise the question in the counterfactual framework proposed by Pearl (2019), we consider the following statements:

  $x'$: The test data is $\mathcal{E}_{\text{test}}$.

  $x$: The test data is a temporally distorted version of $\mathcal{E}_{\text{test}}$.

  $y'$: The performance metric is in the range $(\alpha - \epsilon, \min\{1, \alpha + \epsilon\})$.

  $y$: The performance metric is strictly less than $\alpha - \epsilon$.

Then, the counterfactual question can be framed as $P(y_x \mid x', y')$ which stands for:

> The probability that the prediction accuracy would be less than $\alpha - \epsilon$ had the test data been a temporally distorted version of $\mathcal{E}_{\text{test}}$, given the prediction accuracy was observed to be approximately $\alpha$ when the model was tested on $\mathcal{E}_{\text{test}}$.

We link the counterfactual question to our hypothesis in the following proposition:

**Proposition 3.1.** *If $P(y_x \mid x', y') \approx 0 \implies$ model $f$ cannot learn the temporal patterns in $\mathcal{E}_{\text{train}}$.*

*Proof.*  Consider the set of logical statements:

  $\mathsf{s}_1$: The temporal graph $\mathcal{E}$ contains patterns that allow future edge predictions to be made based on past information, i.e., $\mathcal{G}$ is learnable.

  $\mathsf{s}_2$: The model $f$ is *capable* of learning the patterns in a learnable temporal graph.

  $\mathsf{s}_3$: $\mathcal{E}_{\text{train}} = \mathcal{E}(0, \tau_0), \mathcal{E}_{\text{test}} = \mathcal{E}(\tau_0, T)$.

  $\mathsf{s}_4$: $\mathcal{G}' = \mathscr{D}(\mathcal{E}_{\text{test}})$, where $\mathscr{D}(\cdot)$ is the temporal distortion function.

  $\mathsf{s}_5$: The model $f$ is trained on $\mathcal{E}_{\text{train}}$.

  $\mathsf{s}_6$: The prediction metric reported by $f$ on the real test data $\mathcal{E}_{\text{test}}$ is higher than the prediction metric on the distorted data $\mathcal{G}'$.

$$\mathsf{s}_1 \wedge \mathsf{s}_2 \wedge \mathsf{s}_3 \wedge \mathsf{s}_4 \wedge \mathsf{s}_5 \implies \mathsf{s}_6 \tag{4}$$
$$\neg\mathsf{s}_6 \implies \neg\mathsf{s}_1 \vee \neg\mathsf{s}_2 \vee \neg\mathsf{s}_3 \vee \neg\mathsf{s}_4 \vee \neg\mathsf{s}_5 \tag{contraposition}$$

For the experimental setup $\mathsf{s}_3 = 1$, and $\mathsf{s}_5 = 1$. Assuming that the temporal graph $\mathcal{G}$ is learnable $\mathsf{s}_1 = 1$, and that the function $\mathscr{D}(\mathcal{E}_{\text{test}})$ results in a temporally distorted version of $\mathcal{E}_{\text{test}}$, i.e., $\mathsf{s}_4 = 1$, we get $\neg\mathsf{s}_6 \implies \neg\mathsf{s}_2$. Alternatively, $\neg\mathsf{s}_6 \equiv \mathbb{I}(P(y_x \mid x', y') \approx 0)$, and $\neg\mathsf{s}_2$ is interpreted as "model $f$ is *incapable* of learning the temporal patterns in $\mathcal{G}$".  $\square$

**Example**  In Fig. 1, we show that $\mathcal{E}_{\text{train}} \cup \mathcal{E}_{\text{test}}$ is sampled from a point process with intensity $\lambda^\star(t), t \in [0, T]$. We generate $\mathcal{E}'$ from another point process with intensity $\lambda'(t), t \in [\tau_0, T]$. We depict the intensity functions as two sinusoidal waves with different frequency and phase. If a model $f$ learns this intensity function by observing $\mathcal{E}_{\text{train}}$, and then generates samples for prediction, they would be more similar to $\mathcal{E}_{\text{test}}$ than $\mathcal{E}'$.

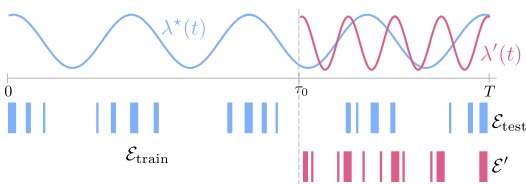

Figure 1: Example of Temporal distortion.

## 3.1 Temporal Distortion Techniques

Let $\mathcal{E}$ be a temporal graph sampled from a temporal point process with intensity $\lambda^\star(t)$ for $t \in [0, T]$. Let $\mathcal{E}'$ be data sampled from another point process with intensity $\lambda'(t)$ for $t \in [0, T]$.

**Definition 3.1.** The temporal graph $\mathcal{E}'$ is $\delta$-**temporally distorted** w.r.t. $\mathcal{E}$ if for some $\delta > 0$,

$$\frac{1}{T} \int_0^T |\lambda^*(t) - \lambda'(t)| \, dt > \delta. \tag{5}$$

In practice, we do not have access to the true intensity functions, and have to compare the realisations instead. Let $\mathcal{E}$ and $\mathcal{E}'$ be two temporal graphs, then we measure the difference in their characteristics through the following two metrics.

**Definition 3.2.** The average time difference (ATD) between $\mathcal{E}$ and $\mathcal{E}'$ is defined as:

$$\mathsf{ATD}(\mathcal{E}, \mathcal{E}') \triangleq \frac{1}{T|\mathcal{E}|} \sum_{(u,v,t) \in \mathcal{E}} \min_{t' \in \mathscr{T}\left(\mathcal{E}'_{(u,v)}\right) \cup \{T\}} |t - t'|, \tag{6}$$

where $T = \max \mathscr{T}(\mathcal{E}) - \min \mathscr{T}(\mathcal{E})$.

**Definition 3.3.** The average count difference (ACD) between $\mathcal{E}$ and $\mathcal{E}'$ is defined as:

$$\mathsf{ACD}(\mathcal{E}, \mathcal{E}') \triangleq \frac{1}{|\mathcal{E}|} \sum_{(u,v,t) \in \mathcal{E}} \left| |\mathcal{E}_{(u,v)}(t - \bar{\tau}, t + \bar{\tau})| - |\mathcal{E}'_{(u,v)}(t - \bar{\tau}, t + \bar{\tau})| \right|, \tag{7}$$

where $\bar{\tau} = \frac{\max \mathscr{T}(\mathcal{E}) - \min \mathscr{T}(\mathcal{E})}{|\mathcal{E}|}$.

Now that we are equipped with metrics to measure the difference between two temporal graphs, we device distortion functions $\mathscr{D}(\cdot)$ which can enable us to investigate the counterfactual question posed earlier. We propose two distortion techniques $\mathscr{D}_{\text{INTENSE}}(\cdot, K)$ which creates $K$ time-perturbed copies of each edge events, and $\mathscr{D}_{\text{SHUFFLE}}(\cdot)$ wherein the timestamps of different edge events are shuffled.

**INTENSE** Let the real temporal graph data be denoted by $\mathcal{E} = \cup_{(u,v) \in \mathcal{U} \times \mathcal{V}} \mathcal{E}_{(u,v)}$, and the distorted version be denoted by $\mathcal{E}' = \cup_{(u,v) \in \mathcal{U} \times \mathcal{V}} \mathcal{E}'_{(u,v)}$. then, for each edge event $(u, v, t)$ in the real data $\mathcal{E}$, we create $K$ edge events $(u, v, t + \tau)$ with $\tau$ sampled uniformly from $(-\bar{\tau}, \bar{\tau})$ for some $\bar{\tau} > 0$. Alternatively, if it is known that $\mathcal{E}_{(u,v)}$ is sampled from a point process with intensity $\lambda^\star_{(u,v)}(t)$, then we can generate $\mathcal{E}'_{(u,v)}$ by sampling from another point process with intensity $\lambda'_{(u,v)}(t)$, such that

$$\lambda'_{(u,v)}(t) = K \lambda^\star_{(u,v)}(t), \forall (u, v) \in \mathcal{U} \times \mathcal{V}.$$

The operation of $\mathscr{D}_{\text{INTENSE}}$ is described in Algorithm 1.

**SHUFFLE** For any two edge events $(u, v, t), (u', v', t') \in \mathcal{E}$, we shuffle the timestamps in the distorted version, i.e. $(u, v, t'), (u', v', t) \in \mathcal{E}'$. The shuffling process is also called label permutation (Chatterjee, 2018). In terms of the point process, we can explain shuffling as follows. If $\mathcal{E}_{(u,v)}$ is known to be sampled from a point process with intensity $\lambda^\star_{(u,v)}(t)$, then $\mathcal{E}'_{(u,v)}$ can be generated by sampling from an inhomogenous Poisson point process with intensity $\lambda'_{(u,v)}(t)$, where

$$\lambda'_{(u,v)}(t) = \frac{\int_0^T \lambda^\star_{(u,v)}(t) \, dt}{\int_0^T \lambda^\star(t) \, dt} \lambda^\star(t), \forall (u, v) \in \mathcal{U} \times \mathcal{V}.$$

We describe the operation of $\mathscr{D}_{\text{SHUFFLE}}$ in Algorithm 2.

---

**Algorithm 1** $\mathscr{D}_{\text{INTENSE}}$

**Input** $\mathcal{E}, K \in \mathbb{N}$
**Output** $\mathcal{E}'$
1: $\mathcal{E}' = \varnothing$
2: $\tau_0 \leftarrow \min \mathscr{T}(\mathcal{E})$
3: $T \leftarrow \max \mathscr{T}(\mathcal{E})$
4: $\bar{\tau} \leftarrow \frac{T - \tau_0}{|\mathcal{E}|}$
5: **for** $(u, v, t) \in \mathcal{E}$ **do**
6:     **for** $k \in [K]$ **do**
7:         $\tau \sim \text{Uniform}(-\bar{\tau}, \bar{\tau})$
8:         $\mathcal{E}' \leftarrow \mathcal{E}' \cup \{(u, v, t + \tau)\}$
9:     **end for**
10: **end for**

---

**Algorithm 2** $\mathscr{D}_{\text{SHUFFLE}}$

**Input** $\mathcal{E}$
**Output** $\mathcal{E}'$
1: $\mathcal{E}' = \varnothing$
2: $\mathcal{T} \leftarrow \mathscr{T}(\mathcal{E})$
3: **for** $(u, v, t) \in \mathcal{E}$ **do**
4:     $\tau \sim \mathcal{T}$
5:     $\mathcal{E}' \leftarrow \mathcal{E}' \cup \{(u, v, \tau)\}$
6:     $\mathcal{T} \leftarrow \mathcal{T} \setminus \{\tau\}$
7: **end for**

---

# 4 Experiment

**Datasets** We use the following datasets[1] to perform counterfactual analysis:

- `wikipedia` (Kumar et al., 2019) describes a dynamic graph of interaction between the editors and Wikipedia pages over a span of one month. The entries consist of the user ID, page ID, and timestamp. The edge features are LIWC-feature vectors (Pennebaker et al., 2001) of the edit text. The edge feature dimension is 172.

- `reddit` (Kumar et al., 2019) describes a bipartite interaction graph between the users and subreddits. The interaction event is recorded with the IDs of the user, subreddit and timestamp. Similar to `wikipedia`, the post content is converted into a LIWC-feature vector of dimension 172 which serves as the edge feature.

- `uci` (Panzarasa et al., 2009) is a dynamic graph describing message-exchange among the students at University of California at Irvine (UCI) from April to October 2004. The interaction event consists of the user IDs, and timestamp.

The scale of the datasets are presented in Table 1. The datasets are chronologically split in the ratio 0.7 : 0.15 : 0.15 into train, validation, and test sets, respectively.

Next, we use $\mathscr{D}_{\text{INTENSE}}(\cdot, 5)$ and $\mathscr{D}_{\text{SHUFFLE}}(\cdot)$ to create 10 temporally distorted samples of the test splits of each dataset. In Table 2, we present the ATD, and ACD by comparing the distorted samples with the original test data of different datasets. Through $\mathscr{D}_{\text{INTENSE}}(\cdot, 5)$, the ATD is negligible, however, the ACD is close to 5. Through $\mathscr{D}_{\text{SHUFFLE}}(\cdot)$, the ACD is approximately 1 for `wikipedia` and `reddit`, and close to 2 for `uci`. We also see an increase in ATD which is close to 0.1 for all datasets. Therefore, the metrics ATD and ACD should be considered in conjunction to measure the dissimilarity of two temporal graphs.

Table 1: Number of nodes and edges in temporal graph datasets.

| Dataset | $|\mathcal{U} \cup \mathcal{V}|$ | $|\mathcal{E}|$ |
|---|---|---|
| wikipedia | 9227 | 157474 |
| reddit | 10984 | 672447 |
| uci | 1899 | 59835 |

Table 2: Distortion measures on different datasets.

| | wikipedia | | reddit | | uci | |
|---|---|---|---|---|---|---|
| | ATD | ACD | ATD | ACD | ATD | ACD |
| INTENSE | 6.9e-6 $\pm$ 2e-8 | 4.479 $\pm$ 1.9e-3 | 1.6e-6 $\pm$ 2e-9 | 4.112 $\pm$ 3.9e-4 | 1.6e-5 $\pm$ 1.2e-7 | 7.214 $\pm$ 1.2e-2 |
| SHUFFLE | 0.078 $\pm$ 5.7e-4 | 1.093 $\pm$ 3.4e-4 | 0.099 $\pm$ 3e-4 | 1.033 $\pm$ 8e-5 | 0.132 $\pm$ 8.4e-4 | 1.877 $\pm$ 3.3e-3 |

**Models** We evaluate[2] the performance of the following TLP models[3] in light of Proposition 3.1:

- `JODIE` (Kumar et al., 2019) uses a recurrent neural network (RNN) to generate node embeddings for each interaction event. The future embedding of a node is estimated through a novel projection operator which is turn in used to predict future edge events.

- `TGAT` (Xu et al., 2020) relies on self-attention mechanism to generate node embeddings to capture the temporal evolution of the graph structure.

- `TGN` (Rossi et al., 2020) combine memory modules with graph-based operators to create an encoder-decoder pair capable of creating temporal node embeddings.

- `CAWN` (Wang et al., 2020) propose a novel strategy based on the law of triadic closure, where temporal walks retrieve the dynamic graph motifs without explicitly counting and selecting the motifs. The node IDs are replaced with the hitting counts to facilitate inductive inference.

For all the models we have forked the main branch of their original Github repositories, and added additional arguments to account for the distortion technique, as well as more focused logging. We wanted to evaluate `GraphMixer` (Cong et al., 2023) as it claims superior performance, however the distorted datasets we generated were not compatible with the dataloader used in their codebase.

---

[1] The datasets can be downloaded from https://zenodo.org/records/7213796

[2] **GPU**: NVIDIA GeForce RTX™ 3060. **CPU**: 12th Gen Intel® Core™ i7-12700 × 20; 16.0 GiB.

[3] The optimal hyper-parameters reported by the models are used.

**Results** The models are evaluated under two settings: *transductive*, and *inductive*. In transductive TLP, the nodes $u, v$ in the positive sample $(u, v, t) \in \mathcal{E}_{\text{test}}$ were observed during training. In contrast, in inductive TLP, at least one node in $u, v$ is novel, and was not observed during training.

Table 3: Performance of the models JODIE, TGAT, TGN, and CAWN on three datasets, and their temporally distorted versions denoted as INTENSE, and SHUFFLE. For each metric, we report the mean, and the 95% confidence interval (CI) as mean $\pm$ CI. We have marked the metrics in blue for distortions that showed that a model was incapable of learning on a certain dataset as per Proposition 3.1, and orange otherwise, with $\epsilon = 0.05$.

| JODIE | wikipedia | | reddit | | uci | |
|---|---|---|---|---|---|---|
| | AU-ROC | AP | AU-ROC | AP | AU-ROC | AP |
| *transductive* | $0.9170 \pm$ 3e-3 | $0.9137 \pm$ 5e-3 | $0.9679 \pm$ 4e-3 | $0.9654 \pm$ 5e-3 | $0.8950 \pm$ 3e-3 | $0.8726 \pm$ 5e-3 |
| INTENSE | $0.9177 \pm$ 7e-3 | $0.9078 \pm$ 1e-2 | $0.9619 \pm$ 9e-3 | $0.9567 \pm$ 1e-2 | $0.9244 \pm$ 2e-3 | $0.9129 \pm$ 5e-3 |
| SHUFFLE | $0.9097 \pm$ 2e-2 | $0.8962 \pm$ 4e-2 | $0.9661 \pm$ 1e-2 | $0.9613 \pm$ 4e-2 | $0.8852 \pm$ 3e-3 | $0.8509 \pm$ 3e-3 |
| *inductive* | $0.8941 \pm$ 4e-3 | $0.8970 \pm$ 5e-3 | $0.9343 \pm$ 9e-3 | $0.9138 \pm$ 2e-2 | $0.7546 \pm$ 8e-3 | $0.7310 \pm$ 2e-2 |
| INTENSE | $0.9036 \pm$ 1e-2 | $0.8972 \pm$ 1e-2 | $0.9457 \pm$ 3e-2 | $0.9308 \pm$ 4e-2 | $0.8384 \pm$ 3e-3 | $0.8332 \pm$ 8e-3 |
| SHUFFLE | $0.9157 \pm$ 1e-2 | $0.9078 \pm$ 2e-2 | $0.9419 \pm$ 3e-2 | $0.9251 \pm$ 6e-3 | $0.7368 \pm$ 5e-3 | $0.6994 \pm$ 8e-3 |

| TGAT | wikipedia | | reddit | | uci | |
|---|---|---|---|---|---|---|
| | AU-ROC | AP | AU-ROC | AP | AU-ROC | AP |
| *transductive* | $0.9499 \pm$ 2e-3 | $0.9528 \pm$ 2e-3 | $0.9806 \pm$ 6e-4 | $0.9818 \pm$ 6e-4 | $0.7885 \pm$ 1e-2 | $0.7694 \pm$ 7e-3 |
| INTENSE | $0.9680 \pm$ 2e-3 | $0.9691 \pm$ 2e-3 | $0.9821 \pm$ 6e-4 | $0.9825 \pm$ 6e-4 | $0.8707 \pm$ 1e-2 | $0.8637 \pm$ 2e-2 |
| SHUFFLE | $0.9492 \pm$ 5e-3 | $0.9532 \pm$ 5e-3 | $0.9814 \pm$ 7e-3 | $0.9826 \pm$ 6e-3 | $0.7719 \pm$ 1e-2 | $0.7336 \pm$ 2e-2 |
| *inductive* | $0.9353 \pm$ 2e-3 | $0.9401 \pm$ 2e-3 | $0.9641 \pm$ 1e-3 | $0.9658 \pm$ 1e-3 | $0.7020 \pm$ 8e-3 | $0.7008 \pm$ 1e-2 |
| INTENSE | $0.9604 \pm$ 2e-3 | $0.9621 \pm$ 2e-3 | $0.9676 \pm$ 8e-4 | $0.9676 \pm$ 1e-3 | $0.8019 \pm$ 2e-2 | $0.8095 \pm$ 2e-2 |
| SHUFFLE | $0.9257 \pm$ 7e-3 | $0.9304 \pm$ 7e-3 | $0.9644 \pm$ 7e-3 | $0.9664 \pm$ 3e-3 | $0.6558 \pm$ 7e-3 | $0.6324 \pm$ 1e-2 |

| TGN | wikipedia | | reddit | | uci | |
|---|---|---|---|---|---|---|
| | AU-ROC | AP | AU-ROC | AP | AU-ROC | AP |
| *transductive* | $0.9370 \pm$ 1e-3 | $0.9472 \pm$ 1e-3 | $0.9545 \pm$ 1e-3 | $0.9578 \pm$ 1e-3 | $0.7826 \pm$ 1e-2 | $0.7975 \pm$ 1e-2 |
| INTENSE | $0.9898 \pm$ 1e-3 | $0.9911 \pm$ 6e-4 | $0.9723 \pm$ 2e-3 | $0.9744 \pm$ 2e-3 | $0.9653 \pm$ 3e-3 | $0.9709 \pm$ 3e-3 |
| SHUFFLE | $0.8310 \pm$ 3e-2 | $0.8487 \pm$ 3e-2 | $0.9533 \pm$ 2e-3 | $0.9563 \pm$ 2e-3 | $0.6722 \pm$ 6e-2 | $0.6520 \pm$ 4e-2 |
| *inductive* | $0.9374 \pm$ 1e-3 | $0.9463 \pm$ 1e-3 | $0.9299 \pm$ 1e-3 | $0.9346 \pm$ 1e-3 | $0.7714 \pm$ 6e-3 | $0.7948 \pm$ 6e-3 |
| INTENSE | $0.9903 \pm$ 1e-3 | $0.9908 \pm$ 6e-4 | $0.9617 \pm$ 3e-3 | $0.9645 \pm$ 3e-3 | $0.9592 \pm$ 3e-3 | $0.9650 \pm$ 2e-3 |
| SHUFFLE | $0.8194 \pm$ 2e-2 | $0.8376 \pm$ 3e-2 | $0.9266 \pm$ 4e-3 | $0.9299 \pm$ 3e-3 | $0.6245 \pm$ 2e-2 | $0.6193 \pm$ 9e-3 |

| CAWN | wikipedia | | reddit | | uci | |
|---|---|---|---|---|---|---|
| | AU-ROC | AP | AU-ROC | AP | AU-ROC | AP |
| *transductive* | $0.9886 \pm$ 1e-4 | $0.9901 \pm$ 1e-4 | $0.9864 \pm$ 4e-3 | $0.9884 \pm$ 3e-3 | $0.9162 \pm$ 9e-4 | $0.9397 \pm$ 8e-4 |
| INTENSE | $0.9977 \pm$ 9e-5 | $0.9975 \pm$ 8e-5 | $0.9931 \pm$ 8e-5 | $0.9942 \pm$ 7e-5 | $0.9848 \pm$ 6e-4 | $0.9889 \pm$ 7e-4 |
| SHUFFLE | $0.9868 \pm$ 3e-4 | $0.9887 \pm$ 3e-4 | $0.9859 \pm$ 6e-4 | $0.9880 \pm$ 2e-3 | $0.8495 \pm$ 7e-3 | $0.8866 \pm$ 2e-3 |
| *inductive* | $0.9877 \pm$ 5e-4 | $0.9896 \pm$ 4e-4 | $0.9833 \pm$ 5e-3 | $0.9859 \pm$ 3e-3 | $0.9052 \pm$ 1e-2 | $0.9273 \pm$ 2e-3 |
| INTENSE | $0.9972 \pm$ 6e-4 | $0.9971 \pm$ 1e-5 | $0.9929 \pm$ 8e-5 | $0.9938 \pm$ 8e-5 | $0.9810 \pm$ 3e-3 | $0.9857 \pm$ 2e-3 |
| SHUFFLE | $0.9876 \pm$ 1e-2 | $0.9896 \pm$ 6e-3 | $0.9826 \pm$ 8e-4 | $0.9851 \pm$ 1e-3 | $0.8383 \pm$ 3e-2 | $0.8783 \pm$ 3e-2 |

From Table 3 it is evident that none of the models are capable of distinguishing between the real data, and data sampled from a five-times more intense version. However, we see that TGN is fairly robust when the timestamps of the test data are shuffled, as its performance worsens the most compared to other models. The performance gap between the real and distorted versions decrease as the dataset size increases (see Table. 1).

In Fig. 2, and Fig. 3 we present the metric gap $\mathbb{E}[y_x - y']$ for $x \sim \mathscr{D}_{\text{INTENSE}}(x', 5)$, and $x \sim \mathscr{D}_{\text{SHUFFLE}}(x')$, respectively, for different models in categorical bar plots grouped by the dataset. We

check whether $\max y_x < \min y'$ in an empirical way by checking if $\mathbb{E}[y_x] + \mathtt{CI} < \mathbb{E}[y'] - \mathtt{CI}' \implies \mathbb{E}[y_x] - \mathbb{E}[y'] < -(\mathtt{CI} + \mathtt{CI}')$. Therefore, we plot $\mathbb{E}[y_x] - \mathbb{E}[y']$ as coloured bars, and $-(\mathtt{CI} + \mathtt{CI}')$ as black diamonds. Moreover, we indicate $\epsilon = 0.05$ as the dashed black line passing through $-0.05$.

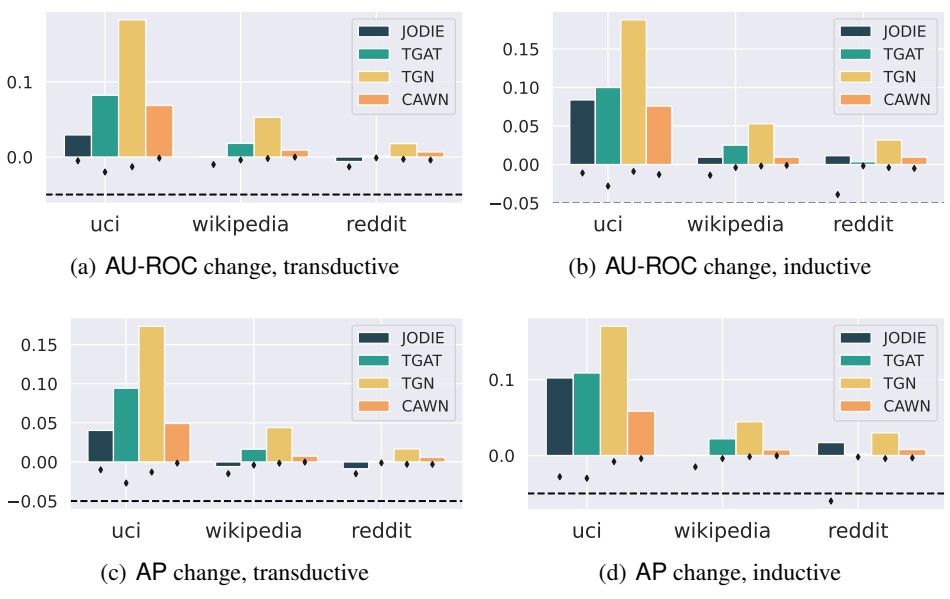

(a) AU-ROC change, transductive

(b) AU-ROC change, inductive

(c) AP change, transductive

(d) AP change, inductive

Figure 2: $\mathbb{E}_{x \sim \mathscr{D}_{\text{INTENSE}}(x', 5)}[y_x - y']$.

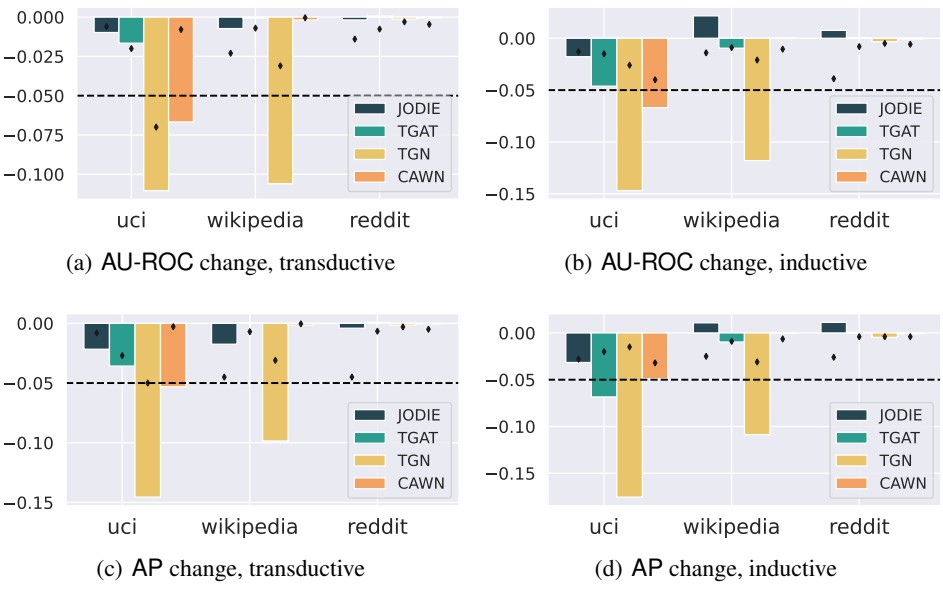

(a) AU-ROC change, transductive

(b) AU-ROC change, inductive

(c) AP change, transductive

(d) AP change, inductive

Figure 3: $\mathbb{E}_{x \sim \mathscr{D}_{\text{SHUFFLE}}(x')}[y_x - y']$.

The models evaluated in this work form the set of baselines to validate the performance of new models. However, as we demonstrate, a higher metric alone is not indicative of good performance without sanity checks. The counterfactual question helps make the evaluation more explainable, as models that perform worse on temporally distorted data with high ATD and ACD can claim superiority over modes that do not. An ideal TLP model should be able to capture the difference in the count of edge events, as well as temporal shifts in the edge events.

# 5 Discussion

Moving away from the binary classification approach to assess the performance of temporal link prediction, the research should explore a generative approach where after observing a temporal graph from time $t \in (0, \tau_0)$, the model can generate a temporal graph in $t \in (\tau_0, T)$. This generated temporal graph should be compared with the ground truth for similarity to assess the performance of the model. The metrics ATD and ACD can be used to measure the difference in the timestamps, as well as the edge counts along the time axis.

We showed that the performance gap in light of Proposition 3.1 decreases with increasing size of the temporal graph, focus should be establish TLP models on smaller datasets, first in the transductive setting, and then progress to inductive setting. In the generative method of evaluation, we can also make use of other metrics that characterise a network, or a point process to add additional constraints.

**Broader Impact**  We presented a framework, wherein we asked a counterfactual question, and then designed intervention mechanisms by generating temporally distorted test sets. In the future, researchers can devise their own temporal distortion techniques to assess the performance of a TLP model, if they follow the binary classification approach to evaluation. Our aim is also to encourage researchers to explore the gnerative evaluation strategy, and design TLP models which can gnerate temporal graphs after observing the edge events in the past. While our work focused on temporal graphs with ephemeral edges (see Definition 2.1), distortion techniques can also be designed for interval graphs, where the edge events persist for a duration. In this work, rather than introducing novel datasets, we present techniques for generating temporally distorted versions of any temporal graph dataset. This makes the contribution relevant even for datasets which will be introduced in the future.

**Limitations**  Due to resource constraints, we could not evaluate the models on more datasets. However, we aim to get additional results by the rebuttal period on the datasets used in Poursafaei and Rabbany (2023). We also wanted to measure the performance of the models through ranking metrics like MRR or MAR, but the distorted datasets were not compatible with the dataloader used by Temporal Graph Benchmark (TGB) (Huang et al., 2024).

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
