# OpenReview forum: "Rethinking Evaluation Strategy for Temporal Link Prediction through Counterfactual Analysis"
_NeurIPS.cc/2024/Datasets_and_Benchmarks_Track — Submitted to NeurIPS 2024 Track Datasets and Benchmarks_

### Official Review · Reviewer_xHrs · 2024-06-18

**Rating:** 5
**Confidence:** 4
**Clarity:** Yes

**Review:**

Pros:

The idea of evaluating the robustness of TLP models by distorted datasets is novel.

Cons:

1.	The authors evaluated robustness of TLP models based on the performance drop on distorted datasets. In specific, the large performance drop, the higher “robustness” of TLP models. However, there are no explanations that why these distorted methods will indeed cause performance drop. In other words, why the TLP models with large performance drop caused by your distortion methods are “good models”?

2.	The proposed distortion strategies, i.e., intense and shuffle, are very intuitive and lack theoretical understandings. Besides, the authors should compare the proposed distortion methods with existing attack methods of (dynamic) graph learning models.
3.	The author only compares 5 baselines and 3 datasets in the experiments, which is insufficient for dynamic graph learning studies, especially for D&B track. Most of these baselines are old (before 2022). The author should compare on other datasets and baselines mentioned in existing papers, for example [1].
4.	This paper lacks related work section. The author should review the existing works of related research fields and discuss the difference to existing works.
5.	The implementations of baseline methods are not integrated with the main repository, making it hard to reproduce the experiments.
[1] Towards Better Dynamic Graph Learning: New Architecture and Unified Library. NeurIPS 2023.
Questions:
1.	In eq. (5), is \lambda (t) for specific node pairs or the whole graph? It needs more explanations.
2.	Why the distorted data cannot be compatible with Graphmixer dataloader?

**Strengths:**

The idea of evaluating the robustness of TLP models by distorted datasets is novel.

**Additional Feedback:**

See reviews above

**Correctness:**

No, The implementations of baseline methods are not integrated with the main repository, making it hard to reproduce the results.

**Documentation:**

No

**Limitations:**

No. I agreed with the authors that more datasets and more evaluation metrics can be included.

**Opportunities For Improvement:**

1.	The ethics of evaluation by “performance drop” is not explained.
2.	The proposed distortion strategies lack theoretical understandings.
3.	The proposed distortion strategies are not compared with existing attacks methods.
4.	The dataset and baselines in experiments are insufficient.
5.	Related work section is missing.
6.	Baseline implementation should be integrated with the main code repository, making it easier to reproduce the results.

**Relation To Prior Work:**

No

**Summary And Contributions:**

This paper evaluates the existing Temporal Link Prediction (TLP) models on temporally distorted test data. The contributions are three-fold. Firstly, they propose a simple techniques to distort temporal graphs. Secondly, they showed the counterfactual performance of TLP models on distorted dataset. Lastly, they propose a novel evaluation strategy for TLP, i.e., comparing the performance drop on the distorted dataset.

---

> ### Author Rebuttal · Authors · 2024-08-16
>
> We thank he reviewer for their invaluable time and effort in providing feedback on our work.
>
> ---
> ## **Explanation of Performance Gap**
> We thank the reviewer for their comments. To improve accessibility, we summarize the main idea in the following paragraph which we will add in the revision:
> > We split the temporal graph chronologically into training and test sets. Then, a model is trained using the training data, and we report the performance metrics on the original test set. Now, we create a temporal distorted version of the test data and also evaluate the model and report the performance metrics. If the performance of the model on the temporally distorted test data is similar or better than the performance on the original test data, then it implies the following:
> the model has not made use of the temporal information in the training set
> or there is no useful temporal information in the dataset
> In the absence of a guarantee that the dataset has temporal information, we can compare different models by comparing the performance gaps.
>
> This is discussed with the help of logical arguments in Sec. 3, and the temporal distortion techniques are explained in Sec 3.1.
>
> ---
> ## **Theoretical Explanation of Distortion Techniques**
> We agree with the reviewer that the distortion techniques are intuitive. We have provided theoretical explanations for the two distortion techniques using point processes. Since we do not deal with the intensity functions of the point process directly, and rather with the realisations, we present the distortion techniques as clear procedures in Algorithms 1 and 2.
>
> Intense:
> >  Alternatively, if it is known that $ \mathcal{E}\_{(u,v)} $ is sampled from a point process with intensity $\lambda\_{(u,v)}^\star(t)$, then we can generate $\mathcal{E}'\_{(u,v)}$ by sampling from another point process with intensity $\lambda'\_{(u,v)}(t)$, such that
> $$ \lambda'\_{(u,v)}(t) = K  \lambda\_{(u,v)}^\star(t), \quad \forall (u,v) \in \mathcal{U} \times \mathcal{V}. $$
>
> Shuffle (we have updated the equation for clarity):
> > If $\mathcal{E}\_{(u,v)}$ is known to be sampled from a point process with intensity $\lambda^\star\_{(u,v)}(t)$, then $\mathcal{E}'\_{(u,v)}$ can be generated by sampling from an  inhomogeneous Poisson point process with intensity $\lambda'\_{(u,v)}(t)$, where
> $$ \lambda'\_{(u,v)}(t) =  \frac{  \left( \int\_{0}^{T} \lambda^\star\_{(u,v)}(t) \, dt \right) \sum\_{(u',v') \in \mathcal{U} \times \mathcal{V}} \lambda^\star\_{(u',v')}(t) }{  \sum\_{(u',v') \in \mathcal{U} \times \mathcal{V}} \int\_{0}^{T} \lambda^\star\_{(u',v')}(t) \, dt}. $$
>
> Although the temporal distortion techniques could be seen as a sort of attack on the data, we would like to highlight that our work is different from adversarial learning. We welcome the reviewer’s suggestion and request them to point us to some temporal graph attacks so we may comment on them more comprehensively, and add that to the discussion.
>
> ---
> ## **More Experiments**
> We thank the reviewer for their comment. We will be presenting additional results on the datasets **mooc** and **lastfm** on two recent TLP baselines `GraphMixer` and `DyGFormer`.
>
> ---
> ## **Related Works**
> We thank the reviewer for this comment. In the paper, we have adopted an approach where we have described the baselines and evaluation techniques in the Introduction to motivate our work. However, we plan to include a Related Works section in the revision to add more context to the counterfactual analysis, with emphasis on the causality in temporal graphs as highlighted by Reviewers **K4d4** and **LMD1**.
>
> ---
> ## **Availibity of Baseline Repositories**
> We thank the reviewer for this comment. In the **README** of the source code, we have explicitly mentioned the forked repositories pointing to the baselines `TGN`, `JODIE`, `TGAT`, and `CAWN`. We have also mentioned that to reproduce the results, one has to download these repositories and run:
> ```
> bash run_all.sh
> ```
>
> We would like to highlight that our contribution is the temporal distortion technique and we do not claim any rights over the baseline code.
>
> ---
> ## **Temporal Distortion Definition**
>
> We thank the reviewer for the attention to detail. We will update equation (5) to the following to improve clarity:
> > $$ \sum\_{(u,v)\in \mathcal{U} \times \mathcal{V}}\frac{1}{T} \int\_{0}^{T} | \lambda\_{(u,v)}^*(t) - \lambda'\_{(u,v)}(t) | \, dt > \delta.$$
>
> ---
> ## **Incompatibility of `GraphMixer` dataloader**
> We could not run `GraphMixer` initially as its data loader is not compatible with the datasets (please see Issue #7 in the official repository on Github for technical details). We also got CUDA compilation issues while running `GraphMixer` (see Issue #8). However, we have checked the code of `DyGLib` introduced to us by Reviewer **zVKR** which consists of `GraphMixer` and `DyGFormer` and is compatible. Therefore, we aim to present additional results on the datasets using these baselines within the discussion period.

---

> > ### Author Rebuttal · Authors · 2024-08-23
> >
> > # Results on Additional Baselines `GraphMixer` and `DyGFormer`
> >
> > |              |              | wikipedia |      | reddit  |         | uci   |          |
> > |--------------|--------------|------------------|------------------|---------------|------------------|-----------------|-----------------|
> > |              |              | AU-ROC |  AP     |  AU-ROC |  AP        |  AU-ROC      |  AP          |
> > | **GraphMixer** | **Transductive** | 0.9654 ± 0.0007  | 0.9690 ± 0.0004  | 0.9727 ± 0.0003 | 0.9738 ± 0.0003  | 0.9176 ± 0.0022  | 0.9323 ± 0.0016  |
> > || **Intense**   | 0.9968 ± 0.0001  | 0.9966 ± 0.0002  | 0.9969 ± 0.0001 | 0.9965 ± 0.0001  | 0.9916 ± 0.0005  | 0.9923 ± 0.0006  |
> > |              | **Shuffle**   | 0.9062 ± 0.0003  | 0.9096 ± 0.0011  | 0.9712 ± 0.0003 | 0.9725 ± 0.0002  | 0.8476 ± 0.0026  | 0.8553 ± 0.0025  |
> > |              | **Inductive** | 0.9600 ± 0.0002  | 0.9639 ± 0.0001  | 0.9489 ± 0.0009 | 0.9517 ± 0.0008  | 0.8960 ± 0.0016  | 0.9133 ± 0.0012  |
> > |              | **Intense**   | 0.9946 ± 0.0001  | 0.9939 ± 0.0001  | 0.9947 ± 0.0002 | 0.9937 ± 0.0002  | 0.9779 ± 0.0001  | 0.9771 ± 0.0005  |
> > |              | **Shuffle**   | 0.8815 ± 0.0023  | 0.8900 ± 0.0023  | 0.9447 ± 0.0011 | 0.9477 ± 0.0007  | 0.7869 ± 0.0003  | 0.7945 ± 0.0003  |
> > | **DyGFormer** | **Transductive** | 0.9890 ± 0.0003  | 0.9901 ± 0.0002  | 0.9913 ± 0.0001 | 0.9921 ± 0.0001  | 0.9478 ± 0.0005  | 0.9596 ± 0.0003  |
> > |              | **Intense**   | 0.9986 ± 0.0001  | 0.9983 ± 0.0001  | 0.9988 ± 0.0001 | 0.9984 ± 0.0001  | 0.9924 ± 0.0001  | 0.9938 ± 0.0001  |
> > |              | **Shuffle**   | 0.9875 ± 0.0001  | 0.9892 ± 0.0001  | 0.9915 ± 0.0001 | 0.9924 ± 0.0001  | 0.9391 ± 0.0008  | 0.9515 ± 0.0012  |
> > |              | **Inductive** | 0.9845 ± 0.0004  | 0.9854 ± 0.0005  | 0.9866 ± 0.0003 | 0.9880 ± 0.0003  | 0.9241 ± 0.0001  | 0.9437 ± 0.0001  |
> > |              | **Intense**   | 0.9976 ± 0.0002  | 0.9965 ± 0.0004  | 0.9981 ± 0.0001 | 0.9973 ± 0.0001  | 0.9831 ± 0.0001  | 0.9854 ± 0.0001  |
> > |              | **Shuffle**   | 0.9812 ± 0.0002  | 0.9833 ± 0.0003  | 0.9866 ± 0.0003 | 0.9878 ± 0.0003  | 0.9057 ± 0.0006  | 0.9291 ± 0.0004  |
> >
> >
> > We are currently obtaining results on the datasets **lastfm** and **mooc** and will report them soon within the rebuttal period.

---

> > ### Author Rebuttal · Authors · 2024-08-26
> >
> > # More Datasets
> >
> > We have now obtained the results for **lastfm** and **mooc**.
> >
> > |  | | **lastfm** | | **mooc** | |
> > |-------------|--------------|------------------|------------------|------------------|------------------|
> > |        |       | AU-ROC      | AP           | AU-ROC       | AP           |
> > | **GraphMixer** | **Transductive** | 0.7406 ± 0.0001  | 0.7630 ± 0.0001  | 0.8363 ± 0.0002  | 0.8233 ± 0.0003  |
> > |             | *Intense*   | 0.9856 ± 0.0001  | 0.9858 ± 0.0001  | 0.9590 ± 0.0001  | 0.9537 ± 0.0001  |
> > |             | *Shuffle*   | 0.7406 ± 0.0001  | 0.7630 ± 0.0001  | 0.8361 ± 0.0002  | 0.8230 ± 0.0002  |
> > |             | **Inductive** | 0.8065 ± 0.0002  | 0.8261 ± 0.0003  | 0.8224 ± 0.0002  | 0.8077 ± 0.0002  |
> > |             | *Intense*   | 0.9864 ± 0.0001  | 0.9867 ± 0.0001  | 0.9592 ± 0.0001  | 0.9555 ± 0.0001  |
> > |             | *Shuffle*   | 0.8065 ± 0.0002  | 0.8261 ± 0.0003  | 0.8222 ± 0.0003  | 0.8072 ± 0.0003  |
> > | **DyGFormer** | **Transductive** | 0.8959 ± 0.0003  | 0.9096 ± 0.0001  | 0.8622 ± 0.0001  | 0.8622 ± 0.0002  |
> > |             | *Intense*   | 0.9911 ± 0.0001  | 0.9912 ± 0.0001  | 0.9728 ± 0.0001  | 0.9709 ± 0.0001  |
> > |             | *Shuffle*   | 0.8959 ± 0.0003  | 0.9096 ± 0.0002  | 0.8622 ± 0.0003  | 0.8620 ± 0.0004  |
> > |             | **Inductive** | 0.9180 ± 0.0002  | 0.9293 ± 0.0001  | 0.8529 ± 0.0002  | 0.8509 ± 0.0003  |
> > |             | *Intense*   | 0.9916 ± 0.0001  | 0.9918 ± 0.0002  | 0.9734 ± 0.0001  | 0.9723 ± 0.0001  |
> > |             | *Shuffle*   | 0.9180 ± 0.0002  | 0.9293 ± 0.0001  | 0.8528 ± 0.0003  | 0.8506 ± 0.0005  |
> > | **CAWN**      | **Transductive** | 0.8494 ± 0.0003  | 0.8755 ± 0.0003  | 0.8653 ± 0.0002  | 0.8667 ± 0.0002  |
> > |             | *Intense*   | 0.9871 ± 0.0001  | 0.9879 ± 0.0002  | 0.9734 ± 0.0001  | 0.9719 ± 0.0001  |
> > |             | *Shuffle*   | 0.8494 ± 0.0003  | 0.8755 ± 0.0003  | 0.8653 ± 0.0004  | 0.8666 ± 0.0003  |
> > |             | **Inductive** | 0.8822 ± 0.0004  | 0.9031 ± 0.0005  | 0.8519 ± 0.0003  | 0.8543 ± 0.0004  |
> > |             | *Intense*   | 0.9882 ± 0.0001  | 0.9889 ± 0.0003  | 0.9737 ± 0.0001  | 0.9731 ± 0.0002  |
> > |             | *Shuffle*   | 0.8822 ± 0.0004  | 0.9030 ± 0.0005  | 0.8518 ± 0.0002  | 0.8541 ± 0.0004  |
> > | **TGAT**      | **Transductive** | 0.7139 ± 0.0004  | 0.7309 ± 0.0003  | 0.8587 ± 0.0002  | 0.8458 ± 0.0003  |
> > |             | *Intense*   | 0.9835 ± 0.0001  | 0.9840 ± 0.0001  | 0.9627 ± 0.0001  | 0.9610 ± 0.0001  |
> > |             | *Shuffle*   | 0.7139 ± 0.0002  | 0.7308 ± 0.0003  | 0.8588 ± 0.0004  | 0.8458 ± 0.0004  |
> > |             | **Inductive** | 0.7661 ± 0.0001  | 0.7817 ± 0.0002  | 0.8563 ± 0.0002  | 0.8430 ± 0.0002  |
> > |             | *Intense*   | 0.9837 ± 0.0002  | 0.9841 ± 0.0001  | 0.9628 ± 0.0001  | 0.9621 ± 0.0001  |
> > |             | *Shuffle*   | 0.7661 ± 0.0002  | 0.7817 ± 0.0002  | 0.8563 ± 0.0002  | 0.8430 ± 0.0003  |

---

> > > ### Comment · Reviewer_xHrs · 2024-08-29
> > >
> > > I thank the authors for their detailed feedbacks. I am glad to see that experiments results on additional datasets and baselines. However, I still do not understand why the “intense” and “shuffle” distortion strategy will cause performance drop. It would be nice if authors could provide theoretical explanations. Considering the authors efforts during rebuttal, I raise my score to 5.

---

> > > > ### Author Rebuttal · Authors · 2024-08-29
> > > >
> > > > # Theoretical Explanation for Performance Drop due to Temporal Distortion Strategies
> > > >
> > > > We thank the reviewer for updating the score. In the following text, we provide further clarification on why the performance of an ideal model must drop if it is presented with temporally distorted data during test time after training on the original data.
> > > >
> > > > ---
> > > >
> > > > The following *text is present in the paper, as well as our first response*. We have explained theoretically, how the intense and shuffle distortions work at the intensity function level (see non-homogeneous Point processes):
> > > > > **Intense:**
> > > > >  Alternatively, if it is known that $ \mathcal{E}\_{(u,v)} $ is sampled from a point process with intensity $\lambda\_{(u,v)}^\star(t)$, then we can generate $\mathcal{E}'\_{(u,v)}$ by sampling from another point process with intensity $\lambda'\_{(u,v)}(t)$, such that
> > > > $$ \lambda'\_{(u,v)}(t) = K  \lambda\_{(u,v)}^\star(t), \quad \forall (u,v) \in \mathcal{U} \times \mathcal{V}. $$
> > > >
> > > > > **Shuffle:**
> > > > > If $\mathcal{E}\_{(u,v)}$ is known to be sampled from a point process with intensity $\lambda^\star\_{(u,v)}(t)$, then $\mathcal{E}'\_{(u,v)}$ can be generated by sampling from an  inhomogeneous Poisson point process with intensity $\lambda'\_{(u,v)}(t)$, where
> > > > $$ \lambda'\_{(u,v)}(t) =  \frac{  \left( \int\_{0}^{T} \lambda^\star\_{(u,v)}(t) \, dt \right) \sum\_{(u',v') \in \mathcal{U} \times \mathcal{V}} \lambda^\star\_{(u',v')}(t) }{  \sum\_{(u',v') \in \mathcal{U} \times \mathcal{V}} \int\_{0}^{T} \lambda^\star\_{(u',v')}(t) \, dt}. $$
> > > >
> > > > An ideal model would predict realizations from $\lambda(t)$, therefore when we compare it with $\lambda'(t)$ (intensity of the distorted version), there will be higher error which would translate to a drop in performance. This concept is explained theoretically through equation 5 in the definition of **$\delta$-temporal distortion**:
> > > > > $$ \sum\_{(u,v)\in \mathcal{U} \times \mathcal{V}}\frac{1}{T} \int\_{0}^{T} | \lambda\_{(u,v)}^*(t) - \lambda'\_{(u,v)}(t) | \, dt > \delta.$$
> > > >
> > > > ---
> > > >
> > > > We trust this response addresses the reviewer’s question. If further clarification is needed, we kindly request the reviewer to specify what qualifies as a theoretical explanation so we can comment accordingly.

---

> > > > > ### Comment · Reviewer_xHrs · 2024-08-30
> > > > >
> > > > > I do not agree that the definition of $\delta$-temporal distortion can be used to quantify the performance drop of model, because $\lambda$ is derived from dataset statistics and unrelated to model performance. The authors might consider using the increase in generalization error due to changes of intensity to measure the extent of performance degradation.

---

> > > > > > ### Author Rebuttal · Authors · 2024-08-31
> > > > > >
> > > > > > We thank the reviewer for their time and for engaging in the discussion.
> > > > > >
> > > > > > We would like to clarify a few things:
> > > > > > - To theoretically justify the counterfactual framework, we have employed a point process model where events on an edge $(u,v)$ are generated via an inhomogeneous Poisson process with intensity $\lambda_{(u,v)}(t)$. The "ideal" model for link prediction in this setting is one which knows the underlying intensities and estimates future edge events based on those. This allows us to talk about how this "ideal" model would perform were its test set distorted.
> > > > > > - The real datasets used in the experiments are, of course, not necessarily generated from a Poisson process, rather we use them to provide a theoretical justification for our procedures. For this reason, they are not statistics of data, rather parameters that (in theory) generated them.
> > > > > > - If the test metrics are not lower on the distorted test set, compared to the original test set, this implies that the model did not make use of the temporal information in the training data. That could be for one of the following reasons: either the dataset has no useful temporal information, or the model is incapable of using temporal information.
> > > > > >
> > > > > >
> > > > > > In Sec. 3.1, we use the example of a dynamic network sampled from an inhomogeneous Poisson process to illustrate how our method elucidates these scenarios. Suppose there is no temporal information, i.e. each $\lambda_{(u,v)}(t)$ is constant for all $t$, then the 'Intense' distortion will introduce spurious events which will result in higher ACD measure. When there is salient temporal information in the data, the 'shuffle' distortion will essentially remove this information by changing the sequence in which the edges occur, and where they occur, resulting in a higher ATD measure. For this reason, if the evaluations metrics tested on this distorted dataset are as good as those on the original data, that suggests that the model is incapable of using the temporal information.
> > > > > >
> > > > > >
> > > > > > **Comment on generalisation gap:**
> > > > > > In all the TLP baselines used, the model with the best performance on the validation set is used, as is a standard practice. While the generalisation gap is a useful metric to see if the model is over or under fitted, it cannot be used by itself to compare two models. *Consider this example*: Let model A have train accuracy of 70% and test accuracy of 50%, and model B has train accuracy of 90% and test accuracy of 60%. The generalisation gap for A is 20, and for B is 30. Then on comparing the generalisation gaps solely would have us believe that model A is better than B while clearly model B is better than model A at the learning task. In contrast, the counterfactual analysis is useful as a sanity check when both the train and test accuracies are high, as is the case for the TLP baselines used in the experiments.

---

### Official Review · Reviewer_K4d4 · 2024-07-23

**Rating:** 6
**Confidence:** 4
**Correctness:** Yes, the claims are correct.
**Clarity:** Yes, the paper is well written.

**Review:**

**originality**: the idea of using counter-factual analysis for link prediction in TG is novel and unexplored before.

**quality**: the paper is well-written and easy to follow. The presented experiments are interesting though still lacking more experiments and settings as is.

**significance**: evaluation on temporal graphs is an important topic. Recent work has studied various aspects and this work proposes a novel aspect with counter-factual analysis, although I still have some concerns see improvements section.

**Strengths:**

pros:

- **novel idea**: evaluating temporal graph learning methods with counter-factual analysis is an interesting and novel idea.

- **clear presentation**: the paper is well written and easy to follow

- **novel metrics**: the proposed ATD and ACD metrics allows the comparison between two temporal graphs, they can also be used in compare similarity between two real temporal graphs in the future.

**Additional Feedback:**

Score raised from 5 to 6.

**Documentation:**

No new datasets are included, the experiments are clearly documented.

**Ethics:**

No ethical concerns.

**Limitations:**

Yes, the authors have adequately addressed the limitations and broader impacts.

**Opportunities For Improvement:**

Overall, I believe this work is novel and interesting, however I have the following concerns and questions which I hope the authors can address:

- **limited experiments**: current empirical experiments are only conducted on three small existing datasets. The authors mentioned the possibility of adding more datasets in the rebuttal period, it will be interesting to see more variations across datasets especially for those that are order of magnitude larger than the ones used especially considering that the performance gap between the distorted and real graph diminishes as graph gets larger.

- **practical considerations**: I have a few concerns regarding applying counter-factual analysis in real world settings. First, the generated distorted temporal graphs are sampled stochastically thus potentially causing reproducibility concerns. Second, how different are the generated distorted graphs, are there some statistics showing the differences between them and the real graph beyond the ATD and ACD metrics proposed in the paper (which might not fully categorize the difference between temporal graphs).

- **current distortion strategies are all time based distortions**: if I understand correctly, only the timestamps are shuffled or distorted in the two proposed distortion strategy. That only perturbs the temporal dimension but not the spatio dimension of the temporal graph. It would be more interesting to see more distortion techniques which adds fake edges or remove edges for example. In addition, temporal distortion also means that for any static graph method, the distorted graph has no difference than the original right?

- if a method performs well on the distorted sample, does it mean it is a bad method or is it just robust to noise in timestamps? Would the distortion idea be better used to understand the robustness of TG methods?

- (minor) How is this work related to studying causal structure on temporal graphs[1]? Shuffling timestamps of edges will destroy the causal structures thus reducing the model's capability to capture causal structure for future link prediction.

[1] Qarkaxhija L, Perri V, Scholtes I. De bruijn goes neural: Causality-aware graph neural networks for time series data on dynamic graphs. InLearning on Graphs Conference 2022 Dec 21 (pp. 51-1). PMLR.

**Relation To Prior Work:**

Yes, this is discussed.

**Summary And Contributions:**

In this work, the authors propose a novel approach to evaluate Temporal Link Prediction (TLP) models, addressing limitations of current evaluation. The proposed method is based on a counterfactual question: "What if a TLP model is tested on a temporally distorted version of the data instead of the real data?" They argue that a model truly capturing temporal patterns should perform worse on distorted data.
Therefore, they introduce simple techniques to distort temporal patterns in graphs, creating temporally distorted test splits applicable to any temporal graph dataset. Then, counterfactual analysis is performed on TLP models to evaluate their ability to capture temporal patterns.
Lastly, they propose an alternative evaluation strategy for TLP, addressing limitations of binary classification and ranking metrics, namely average time difference (ATD) and average count difference (ACD)—to comprehensively measure a model's predictive performance.

---

> ### Author Rebuttal · Authors · 2024-08-16
>
> We thank the reviewer for their time and the constructive feedback on our work.
>
> ---
> ## **More Experiments**
> We thank the reviewer for their comment. We have access to the following datasets which are compatible with the dataloader, and previously reported in EdgeBank. A dataset is considered big if it has a large number of unique edges, and total edges.
>
> In the current work, we have presented results for **uci**, **wikipedia** and **reddit** (arranged from small to large). The datasets **enron**, **SocialEvo**, and **Contact** are smaller than **uci**, while **lastfm** and **mooc** are larger than reddit. Therefore, we will present more results on the datasets **lastfm** and **mooc**.
>
> | Dataset| nodes ($\times 10^3$) | total edges ($\times 10^3$) | unique edges ($\times 10^3$) |
> |---|---|---|---|
> | **wikipedia** | 9.23 | 157.47 | 18.25  |
> | **reddit** | 10.98 | 672.45 | 78.52 |
> | **uci** | 1.89 | 59.84 | 20.29 |
> | **enron** | 0.18 | 125.24 | 3.13 |
> | **lastfm** | 1.98 | 1293.10 | 154.99 |
> | **mooc** | 7.14 | 411.75 | 178.44 |
> | **SocialEvo** | 0.07 | 2099.52 | 4.49 |
> | **Contact** | 0.69 | 2426.28 | 8.06 |
>
> ---
> ## **Stochastic Distortions**
> For each real test data, we create multiple stochastic samples of distorted test data, and perform multiple train-test iterations.
>
> > For each iteration i:
> > - train the model on the train data
> > - test the trained model on the real test data: record metric as a(i)
> > - For each stochastically distorted sample j
> >   - test the trained model on distorted test sample j: record metric as b(i,j)
>
> We report the mean and 95% confidence interval of the metrics to ensure statistical reliability.
>
> ---
> ## **Additional Distortion Measures**
> When comparing two interaction graphs $\mathcal{E}$ and $\mathcal{E}’$, the ATD and ACD metrics answer two essential questions:
> - How far is an edge event in $\mathcal{E}’$ compared to $\mathcal{E}$ ?
> - How many times does an edge event occur in $\mathcal{E}’$ in comparison to $\mathcal{E}$ when counting within the same finite temporal window?
>
> We welcome any metric suggestion that the reviewer might have to compare two interaction graphs.
>
> ---
> ## **Spatial Distortion**
> In an interaction graph, the spatial dimension is not well defined as edges appear and disappear so we cannot see a traditional graph with a structure unless we group the edges together in an interval. In that case, we can apply classic static graph statistics like degree distribution, motif count, centrality, etc. This can be a direction for future work, and we will mention this in the discussion. We thank the reviewer for highlighting this.
>
> We thank the reviewer for their suggestion to perform spatial distortion by adding fake edges and removing edges. In the temporal distortion, we are misplacing edges - adding edges where they don’t belong, and removing them from where they belong. If we understand the suggestion correctly, the reviewer wants to add fake edges that never existed in the graph (novel), and remove edges. The latter is the opposite of Intense, where instead of increasing the intensity of an edge, we decrease it. The suggestion sounds interesting, and we will mention it in the discussion under the heading **Directions for future research**.
>
> To the best of our knowledge, we are the first to apply counterfactual analysis to temporal link prediction. We believe this approach opens the door for the research community to build upon and further refine this work, advancing the field in new and meaningful ways.
>
> If the temporal graph is converted to a static graph by removing all the timestamps then the static versions of the original and temporally distorted temporal graphs are the same. Therefore, any static graph method should produce the same result.
>
> ---
> ## **Robustness to Noise**
> We are proposing making more than just a small perturbation to the timestamps here, we are proposing completely distorting them. For this reason, if a method performs well on the distorted dataset, this doesn’t suggest that it is robust to noise in the timestamps, rather that it isn’t making use of the timestamps at all.
>
> ---
> ## **Causal Structure**
> The reference ‘De Bruijn goes neural’ is interesting and in the same spirit as the baseline CAWN. We notice on page 7 that the authors have swapped the timestamps of the edges, which is similar to the Shuffle distortion technique. However, unlike them, we do not change the training set. Moreover, the evaluation is done through supervised node classification, not temporal link prediction. It is an interesting work and we thank the reviewer for introducing it. We will mention this in the Related Works section, which we will be adding during revision based on the suggestions of some reviewers.

---

> > ### Author Rebuttal · Authors · 2024-08-26
> >
> > # More Datasets
> >
> > We have now obtained the results for **lastfm** and **mooc**.
> >
> > |  | | **lastfm** | | **mooc** | |
> > |-------------|--------------|------------------|------------------|------------------|------------------|
> > |        |       | AU-ROC      | AP           | AU-ROC       | AP           |
> > | **GraphMixer** | **Transductive** | 0.7406 ± 0.0001  | 0.7630 ± 0.0001  | 0.8363 ± 0.0002  | 0.8233 ± 0.0003  |
> > |             | *Intense*   | 0.9856 ± 0.0001  | 0.9858 ± 0.0001  | 0.9590 ± 0.0001  | 0.9537 ± 0.0001  |
> > |             | *Shuffle*   | 0.7406 ± 0.0001  | 0.7630 ± 0.0001  | 0.8361 ± 0.0002  | 0.8230 ± 0.0002  |
> > |             | **Inductive** | 0.8065 ± 0.0002  | 0.8261 ± 0.0003  | 0.8224 ± 0.0002  | 0.8077 ± 0.0002  |
> > |             | *Intense*   | 0.9864 ± 0.0001  | 0.9867 ± 0.0001  | 0.9592 ± 0.0001  | 0.9555 ± 0.0001  |
> > |             | *Shuffle*   | 0.8065 ± 0.0002  | 0.8261 ± 0.0003  | 0.8222 ± 0.0003  | 0.8072 ± 0.0003  |
> > | **DyGFormer** | **Transductive** | 0.8959 ± 0.0003  | 0.9096 ± 0.0001  | 0.8622 ± 0.0001  | 0.8622 ± 0.0002  |
> > |             | *Intense*   | 0.9911 ± 0.0001  | 0.9912 ± 0.0001  | 0.9728 ± 0.0001  | 0.9709 ± 0.0001  |
> > |             | *Shuffle*   | 0.8959 ± 0.0003  | 0.9096 ± 0.0002  | 0.8622 ± 0.0003  | 0.8620 ± 0.0004  |
> > |             | **Inductive** | 0.9180 ± 0.0002  | 0.9293 ± 0.0001  | 0.8529 ± 0.0002  | 0.8509 ± 0.0003  |
> > |             | *Intense*   | 0.9916 ± 0.0001  | 0.9918 ± 0.0002  | 0.9734 ± 0.0001  | 0.9723 ± 0.0001  |
> > |             | *Shuffle*   | 0.9180 ± 0.0002  | 0.9293 ± 0.0001  | 0.8528 ± 0.0003  | 0.8506 ± 0.0005  |
> > | **CAWN**      | **Transductive** | 0.8494 ± 0.0003  | 0.8755 ± 0.0003  | 0.8653 ± 0.0002  | 0.8667 ± 0.0002  |
> > |             | *Intense*   | 0.9871 ± 0.0001  | 0.9879 ± 0.0002  | 0.9734 ± 0.0001  | 0.9719 ± 0.0001  |
> > |             | *Shuffle*   | 0.8494 ± 0.0003  | 0.8755 ± 0.0003  | 0.8653 ± 0.0004  | 0.8666 ± 0.0003  |
> > |             | **Inductive** | 0.8822 ± 0.0004  | 0.9031 ± 0.0005  | 0.8519 ± 0.0003  | 0.8543 ± 0.0004  |
> > |             | *Intense*   | 0.9882 ± 0.0001  | 0.9889 ± 0.0003  | 0.9737 ± 0.0001  | 0.9731 ± 0.0002  |
> > |             | *Shuffle*   | 0.8822 ± 0.0004  | 0.9030 ± 0.0005  | 0.8518 ± 0.0002  | 0.8541 ± 0.0004  |
> > | **TGAT**      | **Transductive** | 0.7139 ± 0.0004  | 0.7309 ± 0.0003  | 0.8587 ± 0.0002  | 0.8458 ± 0.0003  |
> > |             | *Intense*   | 0.9835 ± 0.0001  | 0.9840 ± 0.0001  | 0.9627 ± 0.0001  | 0.9610 ± 0.0001  |
> > |             | *Shuffle*   | 0.7139 ± 0.0002  | 0.7308 ± 0.0003  | 0.8588 ± 0.0004  | 0.8458 ± 0.0004  |
> > |             | **Inductive** | 0.7661 ± 0.0001  | 0.7817 ± 0.0002  | 0.8563 ± 0.0002  | 0.8430 ± 0.0002  |
> > |             | *Intense*   | 0.9837 ± 0.0002  | 0.9841 ± 0.0001  | 0.9628 ± 0.0001  | 0.9621 ± 0.0001  |
> > |             | *Shuffle*   | 0.7661 ± 0.0002  | 0.7817 ± 0.0002  | 0.8563 ± 0.0002  | 0.8430 ± 0.0003  |

---

> > > ### Comment · Reviewer_K4d4 · 2024-08-27
> > > **Response to Authors**
> > >
> > > I thank the authors for their detailed response and addressing most of my concerns. I have raised my score from 5 to 6 and hope the authors will continue to improve upon this work especially for the spatial distortion aspect.

---

> > > > ### Author Rebuttal · Authors · 2024-08-27
> > > >
> > > > We thank the reviewer for their insightful feedback and for updating their score for our work.

---

### Official Review · Reviewer_zVKR · 2024-07-24
**Reviews**

**Rating:** 4
**Confidence:** 3
**Correctness:** Don’t know is correct or not.
**Clarity:** The paper is well written.

**Review:**

This paper addresses the inadequacies in current evaluation methods for Temporal Link Prediction (TLP) models. The authors propose a novel evaluation approach to determine whether TLP models capture temporal patterns in the data.

- The paper is well-written overall.
-  The introduction of counterfactual analysis to evaluate TLP models is novel and addresses a significant gap in the current evaluation strategies.
-  The proposed data distortion techniques can be applied to any temporal graph dataset, making the approach widely useful.
-  The proposed ATD and ACD metrics offer a more comprehensive measure of a model’s predictive performance, which could lead to more accurate assessments.


But they have several weaknesses:
-  **Complexity of Distortion Techniques**: While the distortion methods are innovative, they might add complexity to the evaluation process, since the point process needs to model each edge’s intensity function. The complexity might be really high.

- **Lack of new SOTA methods**: The currently state-of-the-art methods, i.e. GraphMixer[1], and DyGFormer[2], are lacking. You only provided 4 methods before 2020, which can not convince me to say that this is a good evaluation method.  Only three datasets are compared: In the temporal link prediction area, all datasets from EdgeBank[3], i.e. MOOC, LastFM, and Enron, are easy to acquire and have exact same dataset structure. Why do you check your method only on three datasets? You could apply to other datasets easily, and I would like to see further results on other datasets.

- **Typos**: In Section 2.2: "inhomoegenous" should be "inhomogeneous."

- **Definition error**:  In Section 2.1: where is the word “ephemeral edges” from? I never see such a term used to describe temporal edges or temporal links. It appears to be a new or uncommon term. I’m afraid that you made it up.

Furthermore, I don’t know why we tested on a distorted test dataset that could prove the model is good.

**Strengths:**

- The paper is well-written overall.
- The introduction of counterfactual analysis to evaluate TLP models is novel and addresses a significant gap in the current evaluation strategies.
- The proposed data distortion techniques can be applied to any temporal graph dataset, making the approach widely useful.
- The proposed ATD and ACD metrics offer a more comprehensive measure of a model’s predictive performance, which could lead to more accurate assessments.

**Additional Feedback:**

I don’t know why we tested on a distorted test dataset that could prove the model is good. More experiments should be provided.

**Documentation:**

They provide the code link.

**Ethics:**

I’m not suspect there are any ethical concerns.

**Limitations:**

- **Complexity of Distortion Techniques**: While the distortion methods are innovative, they might add complexity to the evaluation process, since the point process needs to model each edge’s intensity function. The complexity might be really high.

- **Lack of new SOTA methods**: The currently state-of-the-art methods, i.e. GraphMixer[1], and DyGFormer[2], are lacking. You only provided 4 methods before 2020, which can not convince me to say that this is a good evaluation method.

- **Only three datasets are compared**: In the temporal link prediction area, all datasets from EdgeBank[3], i.e. MOOC, LastFM, and Enron, are easy to acquire and have exact same dataset structure. Why do you check your method only on three datasets? You could apply to other datasets easily, and I would like to see further results on other datasets.

- **Typos**: In Section 2.2: "inhomoegenous" should be "inhomogeneous."

- **Definition error**:  In Section 2.1: where is the word “ephemeral edges” from? I never see such a term used to describe temporal edges or temporal links. It appears to be a new or uncommon term. I’m afraid that you made it up.


Furthermore, I don’t know why we tested on a distorted test dataset that could prove the model is good.


[1] Do we really need complicated model architectures for temporal networks?, ICLR, 2023

[2] Towards better dynamic graph learning: New architecture and unified library, NIPS, 2023

[3] Towards better evaluation for dynamic link prediction, NIPS, 2022

**Opportunities For Improvement:**

More experiments were provided. More explanation on why we tested on a distorted test dataset could prove the model is good.

**Relation To Prior Work:**

Yes, well discussed how this work differs from previous works.

**Summary And Contributions:**

This paper addresses the inadequacies in current evaluation methods for Temporal Link Prediction (TLP) models. The authors propose a novel evaluation approach to determine whether TLP models capture temporal patterns in the data. Their method involves a sanity check based on a counterfactual question: how would a TLP model perform on a temporally distorted version of the data compared to the real data? The key contributions include:
-  Introducing techniques to distort temporal patterns within a graph, creating temporally distorted test splits for well-known datasets.
-  Conducting counterfactual analysis on various TLP models (JODIE, TGAT, TGN, and CAWN) to evaluate their ability to capture temporal patterns.
-  Proposing an alternative evaluation strategy that avoids the limitations of binary classification and ranking methods, introducing two new metrics: average time difference (ATD) and average count difference (ACD).

---

> ### Author Rebuttal · Authors · 2024-08-16
>
> We thank the reviewer for their invaluable feedback and clarifying questions. We answer them one by one under different headings. We also thank the reviewer for highlighting a typo which we will fix during revision.
>
> ---
> ## **Complexity of Distortion Techniques**
> We thank the reviewer for this question. It is not correct that our method needs to model each edge’s intensity function (please see page 5):
> > In practice, we do not have access to the true intensity functions, so we have to compare the realisations instead.
>
> The distorted datasets are constructed directly from the datasets by either duplicating edge events and perturbing the timestamps (for Intense) or simply shuffling the time labels (for Shuffle). The average runtime to create the distorted test sets of different datasets is presented in the following table:
>
> | Distortion | lastfm | mooc | reddit  | uci | wikipedia |
> |----|---|---|---|---|---|
> | **Intense** $(K=5)$ | 3.4243 s | 1.0830 s | 2.4495 s | 0.2140 s | 0.5596 s |
> | **Shuffle** | 0.1557 s | 0.0524 s | 0.4681 s | 0.0297 s | 0.1098 s |
>
> Theoretically, the computational complexity of the distortion techniques is characterised as follows:
> - $D_{\rm Intense}(\mathcal{E}, K)$:  $\mathcal{O}(K|\mathcal{E}|)$
> - $D_{\rm Shuffle}(\mathcal{E})$: $\mathcal{O}(|\mathcal{E}|)$
>
> In short, both distortion techniques are linear in the total number of edge events in $\mathcal{E}$.
>
> In addition, since the distorted datasets are only used for testing, the model only needs to be trained once using the original dataset, and can then be tested on multiple distorted datasets. Typically, evaluation is much faster than training for these models, and therefore, our method does not add much time complexity to the original evaluation process.
>
> ---
> ## **Lack of new SOTA methods**
> We thank the reviewer for introducing `DyGFormer` to us. The codebase is quite useful and compatible with the data format considered in our study. We could not run GraphMixer initially as its data loader is not compatible with the datasets (please see Issue #7 in the official Github repo). We also got CUDA compilation issues while running `GraphMixer` (see Issue #8). However, we have checked the code of `DyGLib` which consists of `GraphMixer` and `DyGFormer` and is compatible.
>
> Therefore, we will present additional results on the two new models by the end of the discussion period as it is time consuming.
>
> ---
> ## **More Datasets**
> We agree with the reviewer that we can indeed perform the counterfactual analysis using more datasets. We will present additional results on **lastfm** and **mooc** by the end of the discussion period.
>
> ---
> ## **"Ephemeral Edges"**
> The phrase temporal graph is not sufficient to clearly specify a dynamic graph. Therefore, we referred to the edges that exist for a very brief amount of time as ephemeral. However,  as this word may be unfamiliar to some readers, **we will replace ephemeral with instantaneous throughout the paper**. Additionally, we would like to note that we have referred to *interaction graphs* and *unevenly sampled edge sequences* as alternative terms used in the literature to further avoid any potential confusion (page 3).
>
> > In TLP literature, continuous-time temporal graphs with **ephemeral** edges are often considered, where edges represent interaction events between two nodes at a specific point in time. Alternatively, temporal graphs can be defined with edges that appear at a certain time and either persist for a duration (Celikkanat et al., 2024; Farzaneh and Coon, 2023) or accumulate indefinitely. In this work, we focus on the **ephemeral edge temporal graph**, also known as **interaction graphs** (Qin et al., 2024) or **unevenly sampled edge sequence** (Qin and Yeung, 2023).
>
> ---
> ## **Counterfactual Analysis**
> We thank the reviewer for their comments. To improve accessibility, we summarise the main idea in the following paragraph which we will add in the revision:
> > We split the temporal graph chronologically into training and test sets. Then, a model is trained using the training data, and we report the performance metrics on the original test set. Now, we create a temporal distorted version of the test data and also evaluate the model and report the performance metrics. If the performance of the model on the temporally distorted test data is similar or better than the performance on the original test data, then it implies **one** the following:
> > - the model has not made use of the temporal information in the training set,
> > - there is no useful temporal information in the dataset.
> >
> > In the absence of a guarantee that the dataset has temporal information, we can compare different models by comparing the performance gaps. This is discussed with the help of logical arguments in Sec. 3, and the temporal distortion techniques are explained in Sec 3.1.
>
> For the wikipedia and uci datasets, at least one of the methods exhibited degraded test performance on the shuffled dataset (Fig. 3), which suggests that there is indeed useful temporal information in these datasets.

---

> > ### Author Rebuttal · Authors · 2024-08-23
> >
> > # Additional Results on new SoTA methods `GraphMixer` and `DyGFormer`
> >
> > |              |              | wikipedia |      | reddit  |         | uci   |          |
> > |--------------|--------------|------------------|------------------|---------------|------------------|-----------------|-----------------|
> > |              |              | AU-ROC |  AP     |  AU-ROC |  AP        |  AU-ROC      |  AP          |
> > | **GraphMixer** | **Transductive** | 0.9654 ± 0.0007  | 0.9690 ± 0.0004  | 0.9727 ± 0.0003 | 0.9738 ± 0.0003  | 0.9176 ± 0.0022  | 0.9323 ± 0.0016  |
> > || **Intense**   | 0.9968 ± 0.0001  | 0.9966 ± 0.0002  | 0.9969 ± 0.0001 | 0.9965 ± 0.0001  | 0.9916 ± 0.0005  | 0.9923 ± 0.0006  |
> > |              | **Shuffle**   | 0.9062 ± 0.0003  | 0.9096 ± 0.0011  | 0.9712 ± 0.0003 | 0.9725 ± 0.0002  | 0.8476 ± 0.0026  | 0.8553 ± 0.0025  |
> > |              | **Inductive** | 0.9600 ± 0.0002  | 0.9639 ± 0.0001  | 0.9489 ± 0.0009 | 0.9517 ± 0.0008  | 0.8960 ± 0.0016  | 0.9133 ± 0.0012  |
> > |              | **Intense**   | 0.9946 ± 0.0001  | 0.9939 ± 0.0001  | 0.9947 ± 0.0002 | 0.9937 ± 0.0002  | 0.9779 ± 0.0001  | 0.9771 ± 0.0005  |
> > |              | **Shuffle**   | 0.8815 ± 0.0023  | 0.8900 ± 0.0023  | 0.9447 ± 0.0011 | 0.9477 ± 0.0007  | 0.7869 ± 0.0003  | 0.7945 ± 0.0003  |
> > | **DyGFormer** | **Transductive** | 0.9890 ± 0.0003  | 0.9901 ± 0.0002  | 0.9913 ± 0.0001 | 0.9921 ± 0.0001  | 0.9478 ± 0.0005  | 0.9596 ± 0.0003  |
> > |              | **Intense**   | 0.9986 ± 0.0001  | 0.9983 ± 0.0001  | 0.9988 ± 0.0001 | 0.9984 ± 0.0001  | 0.9924 ± 0.0001  | 0.9938 ± 0.0001  |
> > |              | **Shuffle**   | 0.9875 ± 0.0001  | 0.9892 ± 0.0001  | 0.9915 ± 0.0001 | 0.9924 ± 0.0001  | 0.9391 ± 0.0008  | 0.9515 ± 0.0012  |
> > |              | **Inductive** | 0.9845 ± 0.0004  | 0.9854 ± 0.0005  | 0.9866 ± 0.0003 | 0.9880 ± 0.0003  | 0.9241 ± 0.0001  | 0.9437 ± 0.0001  |
> > |              | **Intense**   | 0.9976 ± 0.0002  | 0.9965 ± 0.0004  | 0.9981 ± 0.0001 | 0.9973 ± 0.0001  | 0.9831 ± 0.0001  | 0.9854 ± 0.0001  |
> > |              | **Shuffle**   | 0.9812 ± 0.0002  | 0.9833 ± 0.0003  | 0.9866 ± 0.0003 | 0.9878 ± 0.0003  | 0.9057 ± 0.0006  | 0.9291 ± 0.0004  |
> >
> >
> > We are currently obtaining results on the datasets **lastfm** and **mooc** and will report them soon within the rebuttal period.

---

> > ### Author Rebuttal · Authors · 2024-08-26
> >
> > # More Datasets
> >
> > We have now obtained the results for **lastfm** and **mooc**.
> >
> > |  | | **lastfm** | | **mooc** | |
> > |-------------|--------------|------------------|------------------|------------------|------------------|
> > |        |       | AU-ROC      | AP           | AU-ROC       | AP           |
> > | **GraphMixer** | **Transductive** | 0.7406 ± 0.0001  | 0.7630 ± 0.0001  | 0.8363 ± 0.0002  | 0.8233 ± 0.0003  |
> > |             | *Intense*   | 0.9856 ± 0.0001  | 0.9858 ± 0.0001  | 0.9590 ± 0.0001  | 0.9537 ± 0.0001  |
> > |             | *Shuffle*   | 0.7406 ± 0.0001  | 0.7630 ± 0.0001  | 0.8361 ± 0.0002  | 0.8230 ± 0.0002  |
> > |             | **Inductive** | 0.8065 ± 0.0002  | 0.8261 ± 0.0003  | 0.8224 ± 0.0002  | 0.8077 ± 0.0002  |
> > |             | *Intense*   | 0.9864 ± 0.0001  | 0.9867 ± 0.0001  | 0.9592 ± 0.0001  | 0.9555 ± 0.0001  |
> > |             | *Shuffle*   | 0.8065 ± 0.0002  | 0.8261 ± 0.0003  | 0.8222 ± 0.0003  | 0.8072 ± 0.0003  |
> > | **DyGFormer** | **Transductive** | 0.8959 ± 0.0003  | 0.9096 ± 0.0001  | 0.8622 ± 0.0001  | 0.8622 ± 0.0002  |
> > |             | *Intense*   | 0.9911 ± 0.0001  | 0.9912 ± 0.0001  | 0.9728 ± 0.0001  | 0.9709 ± 0.0001  |
> > |             | *Shuffle*   | 0.8959 ± 0.0003  | 0.9096 ± 0.0002  | 0.8622 ± 0.0003  | 0.8620 ± 0.0004  |
> > |             | **Inductive** | 0.9180 ± 0.0002  | 0.9293 ± 0.0001  | 0.8529 ± 0.0002  | 0.8509 ± 0.0003  |
> > |             | *Intense*   | 0.9916 ± 0.0001  | 0.9918 ± 0.0002  | 0.9734 ± 0.0001  | 0.9723 ± 0.0001  |
> > |             | *Shuffle*   | 0.9180 ± 0.0002  | 0.9293 ± 0.0001  | 0.8528 ± 0.0003  | 0.8506 ± 0.0005  |
> > | **CAWN**      | **Transductive** | 0.8494 ± 0.0003  | 0.8755 ± 0.0003  | 0.8653 ± 0.0002  | 0.8667 ± 0.0002  |
> > |             | *Intense*   | 0.9871 ± 0.0001  | 0.9879 ± 0.0002  | 0.9734 ± 0.0001  | 0.9719 ± 0.0001  |
> > |             | *Shuffle*   | 0.8494 ± 0.0003  | 0.8755 ± 0.0003  | 0.8653 ± 0.0004  | 0.8666 ± 0.0003  |
> > |             | **Inductive** | 0.8822 ± 0.0004  | 0.9031 ± 0.0005  | 0.8519 ± 0.0003  | 0.8543 ± 0.0004  |
> > |             | *Intense*   | 0.9882 ± 0.0001  | 0.9889 ± 0.0003  | 0.9737 ± 0.0001  | 0.9731 ± 0.0002  |
> > |             | *Shuffle*   | 0.8822 ± 0.0004  | 0.9030 ± 0.0005  | 0.8518 ± 0.0002  | 0.8541 ± 0.0004  |
> > | **TGAT**      | **Transductive** | 0.7139 ± 0.0004  | 0.7309 ± 0.0003  | 0.8587 ± 0.0002  | 0.8458 ± 0.0003  |
> > |             | *Intense*   | 0.9835 ± 0.0001  | 0.9840 ± 0.0001  | 0.9627 ± 0.0001  | 0.9610 ± 0.0001  |
> > |             | *Shuffle*   | 0.7139 ± 0.0002  | 0.7308 ± 0.0003  | 0.8588 ± 0.0004  | 0.8458 ± 0.0004  |
> > |             | **Inductive** | 0.7661 ± 0.0001  | 0.7817 ± 0.0002  | 0.8563 ± 0.0002  | 0.8430 ± 0.0002  |
> > |             | *Intense*   | 0.9837 ± 0.0002  | 0.9841 ± 0.0001  | 0.9628 ± 0.0001  | 0.9621 ± 0.0001  |
> > |             | *Shuffle*   | 0.7661 ± 0.0002  | 0.7817 ± 0.0002  | 0.8563 ± 0.0002  | 0.8430 ± 0.0003  |

---

> > > ### Comment · Reviewer_zVKR · 2024-08-31
> > >
> > > Thanks for the response. I would like to raise my score from 4 to 6. Since it seems that SOTA methods have some improvement space on authors' proposed datasets.

---

> > > > ### Author Response · Authors · 2024-08-31
> > > > **Score update from 4 to 6**
> > > >
> > > > We thank the reviewer for reading our response.
> > > >
> > > > We also thank the reviewer for their intention to raise the score from 4 to 6. As only one day is left for the discussion, we kindly request the reviewer to update the score so that it can be reflected on **Openreview**.

---

> > > > > ### Comment · Reviewer_zVKR · 2024-08-31
> > > > >
> > > > > The system cannot support us to modify the final rating now.

---

### Official Review · Reviewer_LMD1 · 2024-07-24
**Acceptable paper with a few improvements**

**Rating:** 6
**Confidence:** 4
**Correctness:** Yes.

**Review:**

The paper discusses issues of the existing TLP methods and proposes new ideas to check these methods on distorted temporal graphs along with using two new metrics for better analysis. A few issues are listed below:

1. The definitions 3.2 and 3.3 are not explained clearly and miss the motivation. In addition, more interpretation is needed to explain the results based on these two metrics. What does high ATD and ACD indicates? Or what does high ATD and low ACD indicates? It would be a lot clearer to show all 4 possible scenarios in a table and discuss what they mean with detailed insights.
2. There are only three datasets used in evaluation and their sizes are moderate at best.

**Strengths:**

1. Novelty of the approach. The paper puts another perspective to the existing TLP methodology. As the paper suggests, TLP is a challenging task and current SOTA models perform almost perfect due to the nature of the existing evaluation setup. However, authors show that a different point of view is crucial for this task.
2. The code is clean and easy to reproduce.

**Additional Feedback:**

None.

**Clarity:**

The paper is well-written, except for the discussion and motivation for the proposed two metrics.

**Documentation:**

Yes.

**Limitations:**

See above.

**Opportunities For Improvement:**

See above.

**Relation To Prior Work:**

One potentially related direction is the study of randomized models in temporal networks, please see the following paper and references therein:

https://epubs.siam.org/doi/10.1137/19M1242252

**Summary And Contributions:**

The paper addresses an important issue about current TLP methods and the evaluation techniques. It mainly proposes 2 key ideas: 1) the distorting method that is applicable for every temporal graph. This is used for sanity checks. 2) It proposes a novel approach (two new metrics) to solve some of the issues of binary classification based TLP methods. In addition, authors do a counterfactual analysis on existing SOTAs and make a comprehensive analysis.

---

> ### Author Rebuttal · Authors · 2024-08-16
>
> We thank the reviewer for their time and effort and for the constructive criticism.
>
> ---
> ## **Explanation of the Metrics**
>
> We sincerely thank the reviewer for their time and effort. We agree that the metrics could be explained better. Therefore, we have now introduced a new figure (please see Fig.1 in the **pdf attached**), and also a table for jointly interpreting the metrics in the four scenarios as suggested by the reviewer (please see Table 1 in **attached pdf**).
>
> We believe the following text will add some clarity to the description of the metrics.
>
> > In $\textsf{ATD}$, we measure the time difference between the edge event $(u,v,t) \in \mathcal{E}$ and the closest $(u,v,t') \in \mathcal{E}'$, reporting the average over all the edge events in $\mathcal{E}$. In Fig. 1 we show two temporal graphs as \textit{impulse trains} with each impulse color coded to represent the edge of the sample 3-node graph. Through $\textsf{ATD}$ we can measure the overall difference in the occurrence of an edge event. However, $\textsf{ATD}$ fails to capture the difference in the frequency with which an edge occurs in the two temporal graphs $\mathcal{E}$ and $\mathcal{E}'$. Therefore, we define average count difference $\textsf{ACD}$ to measure the difference in the frequency with which edges occur in the temporal graph.
>
> > For each edge event $(u,v,t)\in \mathcal{E}$, we count the number of occurrences of $(u,v)$ in the time range $(t-\bar{\tau}, t+\bar{\tau})$ in both $\mathcal{E}$ and $\mathcal{E}'$ and measure the count difference. In Fig. 1 we depict the time interval as a light blue box centered around each edge event in $\mathcal{E}$. For $\bar{\tau} \rightarrow 0$, the search becomes restricted to an infinitesimal time interval, with $\textsf{ACD}(\mathcal{E}, \mathcal{E}')$ approaching $1 - \frac{1}{|\mathcal{E}|}\sum_{(u,v,t) \in \mathcal{E}} \mathbb{I}[ (u,v,t) \in \mathcal{E}'  ]$.
>
> ---
> ## **Larger Datasets**
> We thank the reviewer for their comment. We have access to the following datasets which are compatible with the dataloader, and previously reported in EdgeBank. A dataset is considered big if it has a large number of unique edges, and total edges.
>
> In the current work, we have presented results for **uci**, **wikipedia** and **reddit** (arranged from small to large). The datasets **enron**, **SocialEvo**, and **Contact** are smaller than **uci**, while **lastfm** and **mooc** are larger than reddit. Therefore, we will present more results on the datasets **lastfm** and **mooc** using the current 4 baselines, and 2 additional baselines `GraphMixer` and `DyGFormer`.
>
> | Dataset| nodes ($\times 10^3$) | total edges ($\times 10^3$) | unique edges ($\times 10^3$) |
> |---|---|---|---|
> | **wikipedia** | 9.23 | 157.47 | 18.25  |
> | **reddit** | 10.98 | 672.45 | 78.52 |
> | **uci** | 1.89 | 59.84 | 20.29 |
> | **enron** | 0.18 | 125.24 | 3.13 |
> | **lastfm** | 1.98 | 1293.10 | 154.99 |
> | **mooc** | 7.14 | 411.75 | 178.44 |
> | **SocialEvo** | 0.07 | 2099.52 | 4.49 |
> | **Contact** | 0.69 | 2426.28 | 8.06 |
>
> ---
> ## **Related Works**
> We thank the reviewer for pointing us in this direction. We will add a Related Works Section to the revised version where this paper will prove useful. We will also enhance the discussion and add reference to randomised models in temporal networks which can serve as an alternative to point processes to explain the theoretical foundation for counterfactual analysis.
>
> ---

---

> > ### Comment · Reviewer_LMD1 · 2024-08-20
> >
> > Thanks for the response.

---

> ### Author Rebuttal · Authors · 2024-08-26
>
> # Larger Datasets
>
> We have now obtained the results for **lastfm** and **mooc**.
>
> |  | | **lastfm** | | **mooc** | |
> |-------------|--------------|------------------|------------------|------------------|------------------|
> |        |       | AU-ROC      | AP           | AU-ROC       | AP           |
> | **GraphMixer** | **Transductive** | 0.7406 ± 0.0001  | 0.7630 ± 0.0001  | 0.8363 ± 0.0002  | 0.8233 ± 0.0003  |
> |             | *Intense*   | 0.9856 ± 0.0001  | 0.9858 ± 0.0001  | 0.9590 ± 0.0001  | 0.9537 ± 0.0001  |
> |             | *Shuffle*   | 0.7406 ± 0.0001  | 0.7630 ± 0.0001  | 0.8361 ± 0.0002  | 0.8230 ± 0.0002  |
> |             | **Inductive** | 0.8065 ± 0.0002  | 0.8261 ± 0.0003  | 0.8224 ± 0.0002  | 0.8077 ± 0.0002  |
> |             | *Intense*   | 0.9864 ± 0.0001  | 0.9867 ± 0.0001  | 0.9592 ± 0.0001  | 0.9555 ± 0.0001  |
> |             | *Shuffle*   | 0.8065 ± 0.0002  | 0.8261 ± 0.0003  | 0.8222 ± 0.0003  | 0.8072 ± 0.0003  |
> | **DyGFormer** | **Transductive** | 0.8959 ± 0.0003  | 0.9096 ± 0.0001  | 0.8622 ± 0.0001  | 0.8622 ± 0.0002  |
> |             | *Intense*   | 0.9911 ± 0.0001  | 0.9912 ± 0.0001  | 0.9728 ± 0.0001  | 0.9709 ± 0.0001  |
> |             | *Shuffle*   | 0.8959 ± 0.0003  | 0.9096 ± 0.0002  | 0.8622 ± 0.0003  | 0.8620 ± 0.0004  |
> |             | **Inductive** | 0.9180 ± 0.0002  | 0.9293 ± 0.0001  | 0.8529 ± 0.0002  | 0.8509 ± 0.0003  |
> |             | *Intense*   | 0.9916 ± 0.0001  | 0.9918 ± 0.0002  | 0.9734 ± 0.0001  | 0.9723 ± 0.0001  |
> |             | *Shuffle*   | 0.9180 ± 0.0002  | 0.9293 ± 0.0001  | 0.8528 ± 0.0003  | 0.8506 ± 0.0005  |
> | **CAWN**      | **Transductive** | 0.8494 ± 0.0003  | 0.8755 ± 0.0003  | 0.8653 ± 0.0002  | 0.8667 ± 0.0002  |
> |             | *Intense*   | 0.9871 ± 0.0001  | 0.9879 ± 0.0002  | 0.9734 ± 0.0001  | 0.9719 ± 0.0001  |
> |             | *Shuffle*   | 0.8494 ± 0.0003  | 0.8755 ± 0.0003  | 0.8653 ± 0.0004  | 0.8666 ± 0.0003  |
> |             | **Inductive** | 0.8822 ± 0.0004  | 0.9031 ± 0.0005  | 0.8519 ± 0.0003  | 0.8543 ± 0.0004  |
> |             | *Intense*   | 0.9882 ± 0.0001  | 0.9889 ± 0.0003  | 0.9737 ± 0.0001  | 0.9731 ± 0.0002  |
> |             | *Shuffle*   | 0.8822 ± 0.0004  | 0.9030 ± 0.0005  | 0.8518 ± 0.0002  | 0.8541 ± 0.0004  |
> | **TGAT**      | **Transductive** | 0.7139 ± 0.0004  | 0.7309 ± 0.0003  | 0.8587 ± 0.0002  | 0.8458 ± 0.0003  |
> |             | *Intense*   | 0.9835 ± 0.0001  | 0.9840 ± 0.0001  | 0.9627 ± 0.0001  | 0.9610 ± 0.0001  |
> |             | *Shuffle*   | 0.7139 ± 0.0002  | 0.7308 ± 0.0003  | 0.8588 ± 0.0004  | 0.8458 ± 0.0004  |
> |             | **Inductive** | 0.7661 ± 0.0001  | 0.7817 ± 0.0002  | 0.8563 ± 0.0002  | 0.8430 ± 0.0002  |
> |             | *Intense*   | 0.9837 ± 0.0002  | 0.9841 ± 0.0001  | 0.9628 ± 0.0001  | 0.9621 ± 0.0001  |
> |             | *Shuffle*   | 0.7661 ± 0.0002  | 0.7817 ± 0.0002  | 0.8563 ± 0.0002  | 0.8430 ± 0.0003  |

---

### Author Rebuttal · Authors · 2024-08-28

We would like to thank all of the reviewers for their time and their constructive and insightful comments. We were happy to read that in general, the reviewers found our paper well-written and that our approach is novel and addresses a significant gap in current evaluation strategies.

---
We received multiple comments that we could have expanded our experiments to include more, and larger, datasets, and included more recent baselines. In response, we have conducted additional experiments on the **lastfm**, and **mooc** datasets (two commonly used datasets for temporal link prediction), and the `GraphMixer` and `DyGFormer` baselines, as suggested by the reviewers xHrs and zVKR. The additional results are included in the attached pdf (please see Table 1 and 2).

To aid understanding, we have now introduced an illustration of the metrics ATD and ACD in Fig 1 of the attached pdf. Upon acceptance, we plan to revise the paper to include this figure, as well as the additional results.

---

In our direct responses to reviewers, we have addressed the reviewer’s individual questions and concerns. We believe that our paper is a good fit for NeurIPS D&B track and makes a meaningful contribution to temporal graph learning and evaluation.

---

### Decision · Program_Chairs · 2024-09-26

**Decision:**

Reject

**Comment:**

In this work, the authors propose a new approach to evaluating temporal link prediction (TLP) methods. Their key intuition is that a model truly capturing temporal patterns should perform worse on data with distorted patterns. Building on this idea, the paper introduces (1) a distortion method applicable to any temporal graph and (2) two new evaluation metrics. In addition, based on them, the authors analyze existing TLP methods. The paper effectively highlights the need for a new perspective in the evaluation of TLP methods.

The authors provided strong responses during the rebuttal phase, addressing many concerns raised by the reviewers (including zVKR, who wanted to increase the score but could not due to the system setting).

However, some major concerns remain, including:
- The intuitive and theoretical justification for why the proposed evaluations are effective measures is not clearly presented.
- More distortion strategies could be developed for the proposed evaluations to be applied to static graphs.
- Lack of comparison with SOTA methods.
- Absence of the related work section.

While none of the reviewers are negative about the submission, there is also a lack of enthusiasm. The meta-reviewer believes that, despite its merits, the paper could be improved by addressing the reviewers' suggestions. Thus, the meta-reviewer suggests revising this paper for future conferences instead of accepting it as it is.